# Hydrodynamic model of fish orientation in a channel flow

**Maurizio Porfiri[1,2,3]\*, Peng Zhang[2,3], Sean D Peterson[4]\***

[1]Department of Biomedical Engineering, New York University, New York, United States; [2]Department of Mechanical and Aerospace Engineering, New York University, New York, United States; [3]Center for Urban Science and Progress, New York University Tandon School of Engineering, New York, United States; [4]Mechanical and Mechatronics Engineering Department, University of Waterloo, Waterloo, Canada

**Abstract** For over a century, scientists have sought to understand how fish orient against an incoming flow, even without visual and flow cues. Here, we elucidate a potential hydrodynamic mechanism of rheotaxis through the study of the bidirectional coupling between fish and the surrounding fluid. By modeling a fish as a vortex dipole in an infinite channel with an imposed background flow, we establish a planar dynamical system for the cross-stream coordinate and orientation. The system dynamics captures the existence of a critical flow speed for fish to successfully orient while performing cross-stream, periodic sweeping movements. Model predictions are examined in the context of experimental observations in the literature on the rheotactic behavior of fish deprived of visual and lateral line cues. The crucial role of bidirectional hydrodynamic interactions unveiled by this model points at an overlooked limitation of existing experimental paradigms to study rheotaxis in the laboratory.

## Editor's evaluation

The authors present a simple model of fish swimming in a channel and reacting to the surrounding flow with their lateral line and no other sensory system. They demonstrate that the fish stably orients upstream in certain conditions. Particularly, rheotaxis can emerge even in the absence of sensory feedback, purely as a consequence of passive hydrodynamic interactions in the presence of the walls.

**\*For correspondence:**
mporfiri@nyu.edu (MP);
peterson@uwaterloo.ca (SDP)

**Competing interest:** The authors declare that no competing interests exist.

## Introduction

Swimming animals display a complex behavioral repertoire in response to flows (*Chapman et al., 2011*). Particularly fascinating is the ability of several fish species to orient and swim against an incoming flow, a behavior known as rheotaxis. While intuition may suggest that vision is necessary for fish to determine the direction of the flow, several experimental studies of midwater species swimming in a channel have documented rheotaxis in the dark above a critical flow speed (*Coombs et al., 2020*). When deprived of vision, fish lose the ability to hold station and they may perform sweeping, cross-stream movements from one side of the channel to other (*Bak-Coleman et al., 2013*; *Bak-Coleman and Coombs, 2014*; *Elder and Coombs, 2015*; *Figure 1*).

In addition to vision, fish may rely on an array of compensatory sensory modalities to navigate the flow, which utilizes tactile, proprioceptive, olfactory, electric, kinematic, and hydrodynamic signals (*Montgomery et al., 2000*; *von der Emde, 1999*). For example, fish could sense and actively respond to linear accelerations caused by the surrounding flow using their vestibular system (*Pavlov and Tjurjukov, 1995*). Similarly, with the help of tactile sensors on their body surface, fish could maintain their

**eLife digest** One fascinating and perplexing fact about fish is that they tend to orient themselves and swim against the flow, rather than with it. This phenomenon is called rheotaxis, and it has countless examples, from salmon migrating upstream to lay their eggs to trout drift-foraging in a current. Yet, despite over a century of experimental studies, the mechanisms underlying rheotaxis remain poorly understood. There is general consensus that fish rely on water- and body-motion cues to vision, vestibular, tactile, and other senses. However, several questions remain unanswered, including how blind fish can perform rheotaxis or whether a passive hydrodynamic mechanism can support the phenomenon. One aspect that has been overlooked in studies of rheotaxis is the bidirectional hydrodynamic interaction between the fish and the surrounding flow, that is, how the presence of the fish alters the flow, which, in turn, affects the fish.

To address these open questions about rheotaxis, Porfiri, Zhang and Peterson wanted to develop a mathematical model of fish swimming, one that could help understand the passive hydrodynamic pathway that leads to swimming against a flow. Unlike experiments on live animals, a mathematical model offers the ability to remove cues to certain senses without interfering with animal behavior.

Porfiri, Zhang and Peterson modeled a fish as a pair of vortices located infinitely close to each other, rotating in opposite directions with the same strength. The vortex pair could freely move through an infinitely long channel with an imposed background flow, devoid of all sensory information expect of that accessed through the lateral line. Analyzing the resulting system revealed that there is a critical speed for the background flow above which the fish successfully orients itself against the flow, resulting in rheotaxis. This critical speed depends on the width of the channel the fish is swimming in. Depriving the fish of sensory information received through the lateral line does not preclude rheotaxis, indicating that rheotaxis could emerge in a completely passive manner.

The finding that the critical speed for rheotaxis depends on channel width could improve the design of experiments studying the phenomenon, since this effect could confound experiments where fish are confined in narrow channels. In this vein, Porfiri, Zhang and Peterson's model could assist biologists in designing experiments detailing the multisensory nature of rheotaxis. Evidence of the importance of bidirectional hydrodynamic interactions on fish orientation may also inform modeling research on fish behavior.

---

orientation against a current through momentary contacts with their surroundings (*Arnold, 1969*; *Lyon, 1904*). Several modern studies have unveiled the critical role of the lateral line system, an array of mechanosensory receptors located on the surface of fish body (*Montgomery and Baker, 2020*), in their ability to orient against a current (*Baker and Montgomery, 1999*; *Montgomery et al., 1997*), hinting at a hydrodynamics-based rheotactic mechanism that has not been fully elucidated. When deprived of vision, can fish rely only on lateral line feedback to perform rheotaxis? Is there a possibility for rheotaxis to be achieved through a purely passive hydrodynamic mechanism that does not need any sensing?

Through experiments on zebrafish larvae swimming in a laminar flow in a straight tube, *Oteiza et al., 2017* have recently unveiled an elegant hydrodynamic mechanism for fish to actively perform rheotaxis. Utilizing their mechanosensory lateral line, fish can sense the flow along different parts of their body, which is sufficient for them to deduce local velocity gradients in the flow and adjust their movements accordingly. As further elaborated upon by *Dabiri, 2017*, the insight offered by *Oteiza et al., 2017* is grounded in the fundamental relationship between vorticity and circulation given by the Kelvin-Stokes' theorem, so that fish movements will be informed by local sampling of the vorticity field. While offering an elegant pathway to explain rheotaxis, the framework of *Oteiza et al., 2017* does not include a way for rheotaxis to be performed in the absence of information about the local vorticity field. Several experimental studies have shown that fish can perform rheotaxis even when their lateral line is partially or completed ablated, provided that the flow speed is sufficiently large (*Bak-Coleman et al., 2013*; *Bak-Coleman and Coombs, 2014*; *Baker and Montgomery, 1999*; *Elder and Coombs, 2015*; *Montgomery et al., 1997*; *Oteiza et al., 2017*; *Van Trump and McHenry, 2013*).

Mathematical modeling efforts seeking to clarify the mechanisms underlying rheotaxis are scant (*Burbano-L and Porfiri, 2021*; *Chicoli et al., 2015*; *Colvert and Kanso, 2016*; *Oteiza et al., 2017*),

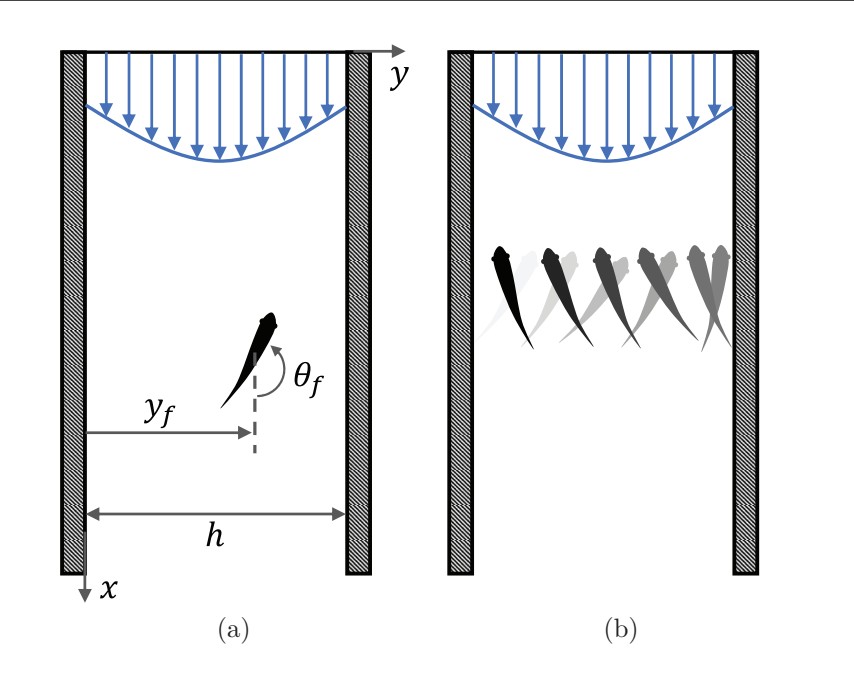

**Figure 1.** Fish rheotaxis. (**a**) Illustration of the problem with notation, showing a fish swimming in a background flow described by *Equation 4*. (**b**) Schematic of the cross-stream sweeping movement of some fish species swimming without visual cues; snapshots of fish at earlier time instants are illustrated by lighter shading.

despite experiments on rheotaxis dating back more than a century (*Lyon, 1904*). A common hypothesis of existing mathematical models is that the presence of the fish does not alter the flow physics with respect to the background flow, thereby neglecting interactions between the fish and the walls of the channel. For example, the model by *Oteiza et al., 2017* implements a random walk in a virtual flow, matching experimental measurements of the background flow in the absence of the animal through particle image velocimetry. A similar line of approach was pursued by *Burbano-L and Porfiri, 2021* for the study of multisensory feedback control of adult zebrafish.

Thus, according to these models, the fish acts as a perfectly non-invasive sensor that probes and reacts to the local flow environment without perturbing it. There are countless examples in fluid mechanics that could question the validity of such an approximation, from coupled interactions between a fluid and a solid in vortex-induced vibrations (*Williamson and Govardhan, 2004*) to laminar boundary layer response to environmental disturbances that range from simple decay of the perturbation to bypass transition (*Saric et al., 2002*). We expect that accounting for bidirectional coupling between the fluid flow and the fish will help clarify many of the puzzling aspects of rheotaxis.

To shed light on the physics of rheotaxis, we formulate a mathematical model based on the paradigm of the finite-dipole, originally proposed by *Tchieu et al., 2012*. Within this paradigm, a fish is viewed as a pair of point vortices of equal and opposite strength separated by a constant distance in a two-dimensional plane. The application of the finite-dipole has bestowed important theoretical advancements in the study of hydrodynamic interactions between swimming animals (*Gazzola et al., 2016*; *Filella et al., 2018*; *Kanso and Cheng Hou Tsang, 2014*; *Kanso and Michelin, 2019*; *Porfiri et al., 2021*), although numerical validation of the framework against full solution of Navier-Stokes equations is lacking – conducting such a validation is also part of this study. Upon validating the dipole model, we investigate the bidirectional coupling between a fish and the surrounding fluid flow in a channel. Our work contributes to the recent literature on minimal models of fish swimming (*Gazzola et al., 2014*; *Gazzola et al., 2015*; *Sánchez-Rodríguez et al., 2020*) that builds on seminal work by *Lighthill, 1975*, *Taylor, 1997* and *Wu, 2006* to elucidate the fundamental physical underpinnings of locomotion and inform the design of engineering systems.

We focus on an ideal condition, where fish are deprived of all sensing systems, other than the lateral line that gives them access to information about the flow. Such flow information is coupled,

however, to the motion of the fish itself, which acts as an invasive sensor and perturbs the background flow. Just as fish motion influences the local flow field, so too does the local flow field alter fish motion through advection. Predictions from the proposed model are compared against existing empirical observations on fish rheotaxis, compiled through a comprehensive literature review of published work since 1900. Data presented in the literature are used to offer context to the predicted dependence of rheotaxis performance on local flow characteristics, individual fish traits, and lateral line feedback.

## Results

### Model of the fluid flow

Consider a single fish swimming in an infinitely long two-dimensional channel of width $h$ (*Figure 1(a)*). Let one wall of the channel be at $y = 0$ and the other at $y = h$, with $x$ pointing along the channel. The fish position at time $t$ is given by $\vec{r}_f(t) = x_f(t)\hat{i} + y_f(t)\hat{j}$, where $\hat{i}$ and $\hat{j}$ are the unit vectors in the $x$ and $y$ directions, respectively. The orientation of the fish with respect to the $x$ axis is given by $\theta_f(t)$ (positive counter-clockwise) and its self-propulsion velocity is $\vec{v}_f = v_0(\cos\theta_f\hat{i} + \sin\theta_f\hat{j}) = v_0\hat{v}_f$, where $v_0$ is the constant speed of the fish and $\hat{v}_f$ is a unit vector in the swimming direction.

The flow is modeled as a potential flow, which is a close approximation of the realistic flow field around a fish. This simple linear fluid model is intended to capture the mean flow physics, thereby averaging any turbulence contribution. The fish is modeled as a dipole, the potential field of which at some location $\vec{r} = x\hat{i} + y\hat{j}$ is given by

$$\phi_f(\vec{r}, \vec{r}_f, \theta_f) = -r_0^2\left(\frac{(\vec{r}-\vec{r}_f)\cdot\vec{v}_f}{\|\vec{r}-\vec{r}_f\|^2}\right),\tag{1}$$

where $r_0$ is the characteristic dipole length-scale (on the order of the amplitude of the fish tail beating), so that the circulation of each vortex is $2\pi r_0 v_0$. This potential field is constructed assuming a far-field view of the dipole (*Filella et al., 2018*), wherein $r_0$ is small in comparison with the characteristic flow length scale, which is satisfied for $\rho = r_0/h \ll 1$. The velocity field at $\vec{r}$ due to the dipole (fish) is $\vec{u}_f = \nabla\phi_f$.

A major contribution of the proposed model is the treatment of the fish as an invasive sensor that both reacts to and influences the background flow, thereby establishing a coupled interaction between the fish and the surrounding environment. A fish swimming in the vicinity of a wall will induce rotational flow near the boundary. In the inviscid limit, this boundary layer is infinitesimally thin and can be considered as wall-bounded vorticity (*Batchelor, 2000*). Employing the classical method of images (*Newton, 2011*), the influence of the wall-bounded vorticity on the flow field is equivalent to that of a fictitious fish (dipole) mirrored about the wall plane. For the case of a fish in a channel, this results in an infinite number of image fish (dipoles) (*Figure 2*), the position vectors for which are

$$\vec{r}^{+}_{<,n} = x_f\hat{i} + (y_f - 2(n+1)h)\hat{j},\tag{2a}$$

$$\vec{r}^{-}_{<,n} = x_f\hat{i} + (-y_f - 2nh)\hat{j},\tag{2b}$$

$$\vec{r}^{+}_{>,n} = x_f\hat{i} + (y_f + 2(n+1)h)\hat{j},\tag{2c}$$

$$\vec{r}^{-}_{>,n} = x_f\hat{i} + (-y_f + 2(n+1)h)\hat{j},\tag{2d}$$

where $n$ is a non-negative integer representing the $n$-th set of images. Subscripts "<" and ">" correspond to position vectors of the images at $y < 0$ and $y > h$, respectively. Likewise, superscript "±" denotes the orientation of the image dipole as $\pm\theta_f$; that is, a position vector with superscript "+" indicates that the associated image has the same orientation as the fish.

The potential function for a given image is found by replacing $\vec{r}_f$ in *Equation 1* with its position vector from *Equation 2d* and adjusting the sign of $\theta_f$ in *Equation 1* to match the superscript of its vector. The potential field at $\vec{r}$ due to the image dipoles is

$$\phi_w(\vec{r}, \vec{r}_f, \theta_f) = \sum_{n=0}^{\infty}\left(\phi_f(\vec{r}, \vec{r}^{+}_{<,n}, \theta_f) + \phi_f(\vec{r}, \vec{r}^{-}_{<,n}, -\theta_f) + \phi_f(\vec{r}, \vec{r}^{+}_{>,n}, \theta_f) + \phi_f(\vec{r}, \vec{r}^{-}_{>,n}, -\theta_f)\right).\tag{3}$$

Thus, the velocity field due to the wall is computed as $\vec{u}_w = \nabla\phi_w$, and the overall velocity field induced by the fish is $\vec{u}_f + \vec{u}_w$. (A closed-form expression for the series in terms of trigonometric and hyperbolic functions is presented in Appendix 1) Overall, the presence of the walls distorts the flow

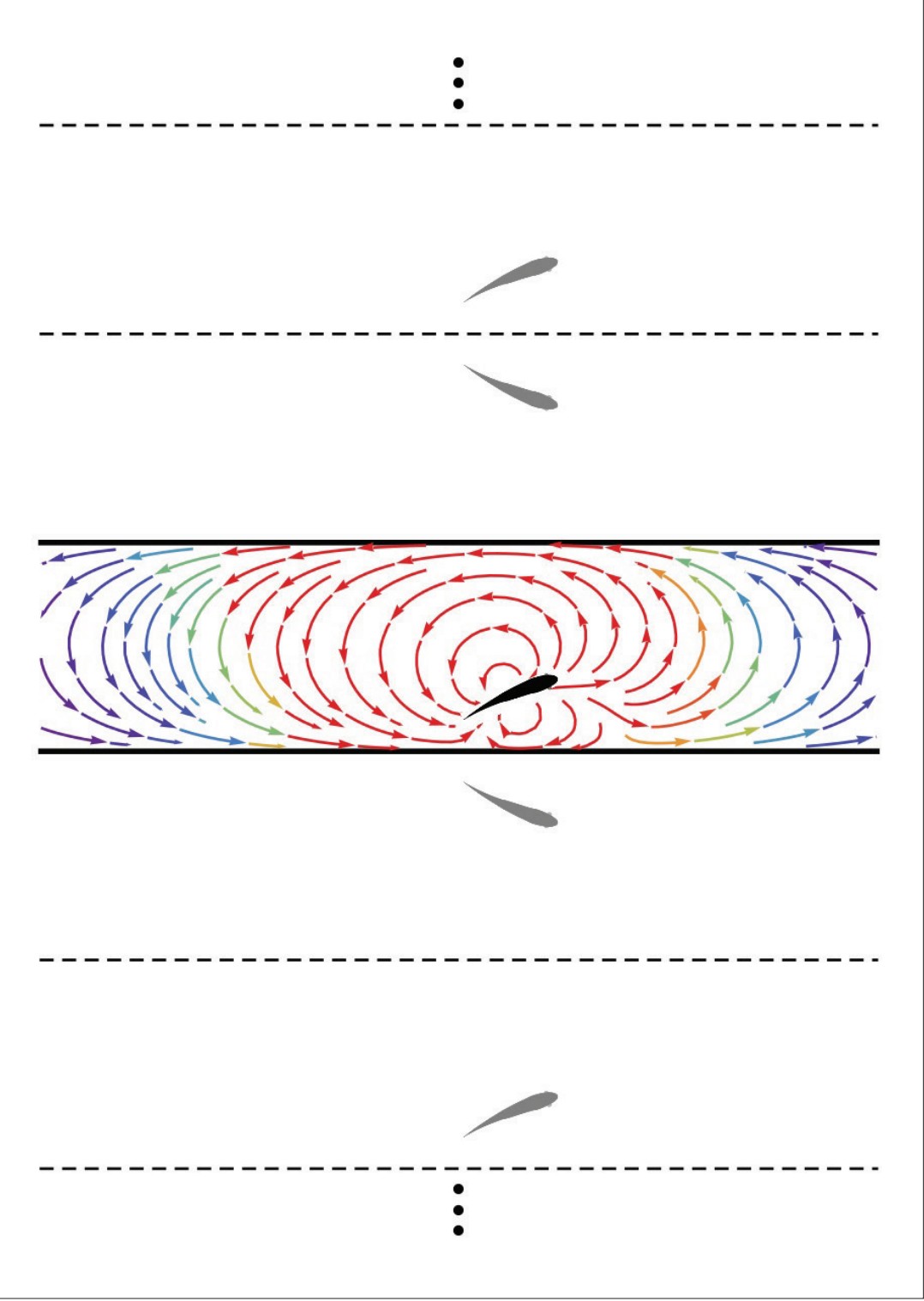

**Figure 2.** Method of images. Schematic of the fish (black) in the channel (thick lines) and the set of images (gray) needed to generate the channel. The streamlines generated by the fish in an otherwise quiescent fluid are shown in the channel colored by local velocity magnitude (red: high; blue: low). Dashed and solid lines are mirroring planes for the method of images, the pattern for which continues *ad infinitum.*

generated by the dipole, both compressing the streamlines between the fish and the walls in its proximity and creating long-range swirling patterns in the channel (**Figure 2**).

The presence of a background flow in the channel is modelled by superimposing a weakly rotational flow,

$$\vec{u}_b(\vec{r}) = U_0 \left( 1 - 4\epsilon \left( \frac{y}{h} - \frac{1}{2} \right)^2 \right) \hat{i}, \tag{4}$$

which has speed $U_0$ at the channel centerline and $U_0(1 - \epsilon)$ at the walls, $\epsilon$ being a small positive parameter. As $\epsilon \to 0$, a uniform (irrotational) background flow is recovered: such a flow is indistinguishable from the one in **Figure 2**, provided that the observer is moving with the background flow.

For $\epsilon \ll 1$, the imposed velocity profile approximates that of a turbulent channel flow, wherein a modest degree of velocity profile curvature is present near the channel centerline. We note that this velocity profile does not satisfy the no-slip boundary condition (zero velocity on the walls), and the flow is entirely described by only two parameters ($U_0$ and $\epsilon$). For $\epsilon \simeq 1$, the profile approaches that of a laminar flow with parabolic dependence on the cross-stream coordinate. The overall fluid flow in the channel is ultimately computed as $\vec{u} = \vec{u}_f + \vec{u}_w + \vec{u}_b$.

The circulation in a region $\mathcal{R}$ in the flow field centered at some location $y$ is $\Gamma = \int_{\mathcal{R}} \omega \, dA$, where $\omega = (\nabla \times \vec{u}) \cdot \hat{k}$ is the local fluid vorticity ($\hat{k} = \hat{i} \times \hat{j}$). For the considered flow field, we determine

$$\omega(\vec{r}) = \frac{8U_0\epsilon}{h} \left( \frac{y}{h} - \frac{1}{2} \right), \tag{5}$$

whereby the irrotational component of the flow field does not contribute to the circulation, and the circulation at a point (per unit area) is equivalent to the local vorticity.

## Numerical validation of the dipole model

Despite the success of dipole-based models in the study of fish swimming (**Filella et al., 2018**; **Gazzola et al., 2016**; **Porfiri et al., 2021**; **Tchieu et al., 2012**), their accuracy against complete Navier-Stokes simulations remains elusive. The potential flow framework in which these models are grounded

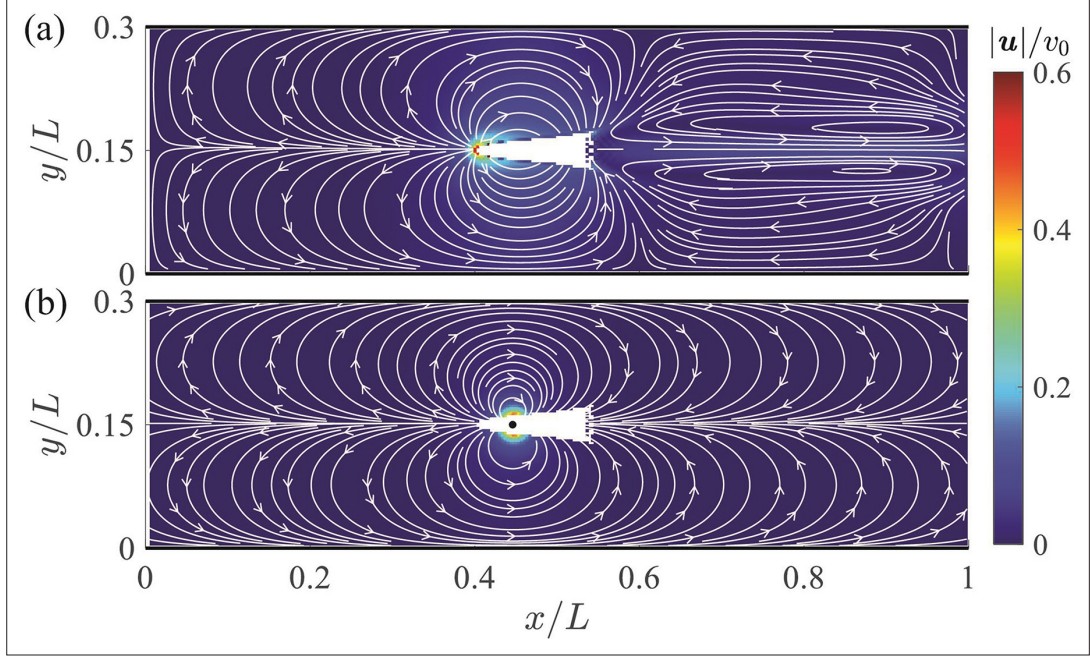

**Figure 3.** Velocity field around a swimming fish from computational fluid dynamics. (**a**) Mean velocity field around the steady swimming giant danio relative to the background flow. (**b**) Velocity field predicted by a dipole with $\theta = \pi$ located at $0.315l$ from the fish head along its centerline relative to the background flow. The selection of the dipole location and strength is detailed in Appendix 2.

neglects boundary layers and the resulting wakes that emerge from viscous effects. Quantifying the extent to which these effects influence the flow field generated by the fish is part of this study.

Specifically, computational fluid dynamics (CFD) simulations were conducted to detail the flow field around a fish during steady swimming. The simulation setup was based upon a giant danio of body length $l = 7.3$ cm in a channel of width $h = 15.0$ cm. The length of the simulation domain was $L = 50.0$ cm ($\sim 6.8l$) with the fish model placed at the channel centerline 20 cm ($\sim 2.7l$) downstream of the inlet. The body undulation of the giant danio was imposed a priori in the simulation, based on data from *Najafi and Abtahi, 2022*. The time-resolved flow field around the fish was quantified by solving the incompressible Navier-Stokes equations. Details on the setup of the numerical framework and convergence analysis supporting its accuracy are included in Appendix 2.

The mean velocity field averaged over a tail beating cycle is displayed in *Figure 3(a)*. The predominant flow feature observed in the mean field is a flow circulation from the head to the tail of the fish with left-right symmetry and compression of the streamlines near the channel walls. The highest velocity is found at the head of the fish with a thrust wake and recirculation region downsteam of the animal – both at a lower velocity than the anterior flow. The largest velocity in the wake is less than 20% of the peak values recorded ahead of the fish. Additional simulation results are included in Appendix 2. The flow field predicted by the dipole model is in good agreement with numerical simulations, as shown in *Figure 3*. The dipole model is successful in capturing the circulation from the head to the tail and the compression of the streamlines near the walls. These features are expected to be the main drivers of the interaction between the fish and the channel walls, thereby supporting the value of a dipole model for a first-order analysis of the hydrodynamics of rheotaxis.

## Model of fish dynamics

From knowledge of the fluid flow in the channel, we compute the advective velocity $\vec{U}(\vec{r}_f, \theta_f)$ and hydrodynamic turn rate $\Omega(\vec{r}_f, \theta_f)$ at the fish location, which encode the influence of the confining walls and background flow on the translational and rotational motion of the fish, respectively. Neglecting the inertia of the fish so that it instantaneously responds to changes in the fluid flow, we determine (*Filella et al., 2018*)

$$\dot{\vec{r}}_f(t) = \vec{U}(\vec{r}_f(t), \theta_f(t)) + \vec{v}_f(\theta_f(t)), \tag{6a}$$

$$\dot{\theta}_f(t) = \Omega(\vec{r}_f(t), \theta_f(t)) + \lambda(\vec{r}_f(t), \theta_f(t)), \tag{6b}$$

where $\lambda$ is the feedback mechanism based on the circulation measurement through the lateral line.

The advective velocity is found by de-singularizing the total velocity field $\vec{u}$ at $\vec{r} = \vec{r}_f$, which is equivalent to calculating the sum of the velocity due to the walls and the background flow in correspondence of the fish (*Milne-Thomson, 1996*)

$$\begin{aligned}
\vec{U}(\vec{r}_f, \theta_f) &= \left( \vec{u}_w(\vec{r}, \vec{r}_f, \theta_f) + \vec{u}_b(\vec{r}) \right) |_{\vec{r}=\vec{r}_f} \\
&= -\frac{\pi^2 v_0 \rho^2}{12} \left[ \left( 1 + 3 \csc^2 \left( \frac{\pi y_f}{h} \right) \right) \cos \theta_f \hat{i} - \left( 1 - 3 \csc^2 \left( \frac{\pi y_f}{h} \right) \right) \sin \theta_f \hat{j} \right] \\
&\quad + U_0 \left( 1 - 4\epsilon \left( \frac{y_f}{h} - \frac{1}{2} \right)^2 \right) \hat{i}.
\end{aligned} \tag{7}$$

*Equation (7)* indicates that the walls have a retarding effect on the swimming speed of the fish that increases in magnitude the closer the fish gets to either wall of the channel. A fish swimming with orientation $\theta_f = 0$ at the center of the channel, for example, will swim with velocity $\dot{\vec{r}}_f(t) = v_0 (1 - (\pi^2/3)\rho^2)\hat{i} + U_0 \hat{i}$. This effect should not be mistaken as traditional viscous drag, which is not included in potential flow theory; rather, it should be intended as the impact of nearby solid boundaries.

Hydrodynamic turn rate is incorporated by considering the difference in velocity experienced by the two constituent vortices comprising the dipole, namely,

$$\begin{aligned}
\Omega(\vec{r}_f, \theta_f) &= -\hat{v}_f \cdot \left[ \nabla \left( \vec{u}_w(\vec{r}, \vec{r}_f, \theta_f) + \vec{u}_b(\vec{r}) \right) |_{\vec{r}=\vec{r}_f} \right] \hat{v}_f^{\perp} \\
&= -\frac{\pi^3 \rho^2 v_0}{4h} \cot \left( \frac{\pi y_f}{h} \right) \csc^2 \left( \frac{\pi y_f}{h} \right) \cos \theta_f + \frac{8 U_0 \epsilon}{h} \left( \frac{y_f}{h} - \frac{1}{2} \right) \cos^2 \theta_f,
\end{aligned} \tag{8}$$

where $\hat{v}_f^\perp = \hat{k} \times \hat{v}_f$; see Materials and methods section for the mathematical derivation. *Equation (8)* indicates that interaction with the walls causes the fish to turn towards the nearest wall; for example, a fish at $y_f = 3/4h$, will experience a turn rate due to the wall of $(\pi^3 \rho^2 v_0)/(2h) \cos\theta_f$, such that it will be rotated counter-clockwise if swimming downstream and clockwise if swimming upstream. On the other hand, the turning direction imposed by the background flow is always positive (counter-clockwise) in the right half of the channel and negative (clockwise) in the left half, irrespective of fish orientation, so that a fish at $y_f = 3/4h$ will always be rotated counter-clockwise. As a result, the fish may turn towards or away from a wall, depending on model parameters and orientation.

Based on experimental observations and theoretical insight (*Burbano-L and Porfiri, 2021*; *Oteiza et al., 2017*), we hypothesize that hydrodynamic feedback, that is, lateral line measurements of the surrounding fluid that fish can employ to navigate the flow, is related to the measurement of the circulation in a region surrounding the fish. This hypothesis is supported by experimental evidence presented in *Oteiza et al., 2017*, which indicated the ability of fish to sense variations in the local velocity gradients to perform rheotaxis. Therein, the authors also found that partial ablation of the lateral line leads to the loss of the rheotactic behavior, hinting at the importance of the flow information on both sides of the fish body for the estimation of the flow circulation.[11]1 Temporal fluctuations, as in turbulent flows, are neglected in our model. As such, we assume time-averaged circulation sensing from the lateral line. We consider a rectangular region $\mathcal{R}$ of width $r_0$ along the fish body length $l$. For simplicity, we assume a linear feedback mechanism, $\lambda(\vec{r}_f, \theta_f) = K\Gamma(\vec{r}_f, \theta_f)$, where we made evident that circulation is computed about the fish location and $K$ is a non-negative feedback gain. Assuming that the fish size is smaller than the characteristic length scale of the flow, we linearize the vorticity along the fish in *Equation 5* as $\omega(\vec{r}) \approx \omega(\vec{r}_f) + \nabla\omega(\vec{r}_f) \cdot \hat{v}_f \Delta l$. By computing the integral from $\Delta l = -l/2$ to $l/2$, we obtain

$$\lambda(\vec{r}_f, \theta_f) = Kr_0 l \frac{8U_0\epsilon}{h}\left(\frac{y_f}{h} - \frac{1}{2}\right). \tag{9}$$

Compared to established practice for modeling fish behavior in response to visual stimuli (*Calovi et al., 2014*; *Couzin et al., 2005*; *Gautrais et al., 2009*; *Zienkiewicz et al., 2015*), the proposed model introduces nonlinear dynamics arising from the bidirectional coupling between the motion of the fish and the flow physics in its surroundings. We note that the employed feedback in *Equation 9* neglects additional potential sensing mechanisms, including vision (*Lyon, 1904*), acceleration sensing through the vestibular system (*Pavlov and Tjurjukov, 1995*), and pressure sensing through sensory afferents in the fins (*Hardy et al., 2016*), which might enhance the ability of fish to navigate the flow.

## Analysis of the planar dynamical system

Given that the right hand side of equation set *Equation 6a* and *Equation 6b* is independent of the streamwise position of the fish, the equations for the cross-streamwise motion and the swimming direction can be separately studied, leading to an elegant nonlinear planar dynamical system. We center the cross-stream coordinate about the center of the channel and non-dimensionalize it with respect to $h$, introducing $\xi = y_f/h - 1/2$. The governing equations become

$$\dot{\xi} = \left[1 - \frac{\pi^2\rho^2}{12}\left(3\csc^2\left(\pi\left(\xi + \frac{1}{2}\right)\right) - 1\right)\right]\sin\theta_f, \tag{10a}$$

$$\dot{\theta}_f = -\frac{\pi^3\rho^2}{4}\cot\left(\pi\left(\xi + \frac{1}{2}\right)\right)\csc^2\left(\pi\left(\xi + \frac{1}{2}\right)\right)\cos\theta_f + 8\alpha\xi\left(\cos^2\theta_f + \kappa\right), \tag{10b}$$

where we non-dimensionalized by the time needed for the fish to traverse the channel in the absence of a background flow, that is, $h/v_0$, and introduced $\alpha = U_0\epsilon/v_0$ and $\kappa = Kr_0 l$ (see Materials and methods section for estimation of these parameters from experimental observations).

In search of the equilibria of the dynamical system, we note that swimming downstream or upstream ($\theta_f = 0$ and $\pi$, respectively) solves *Equation 10a* for any choice of the cross-stream coordinate, the value of which is determined from the solution of *Equation 10b* for the corresponding orientation $\theta_f$. In the case of downstream swimming, the only solution of the resulting transcendental equation is $\xi = 0$. For upstream swimming, depending on the value of the parameter $\beta = (\alpha(1 + \kappa))/\rho^2$, we have one or three solutions: if $\beta < \beta^* = \pi^4/32$, the only solution is $\xi = 0$, otherwise, in addition to $\xi = 0$,

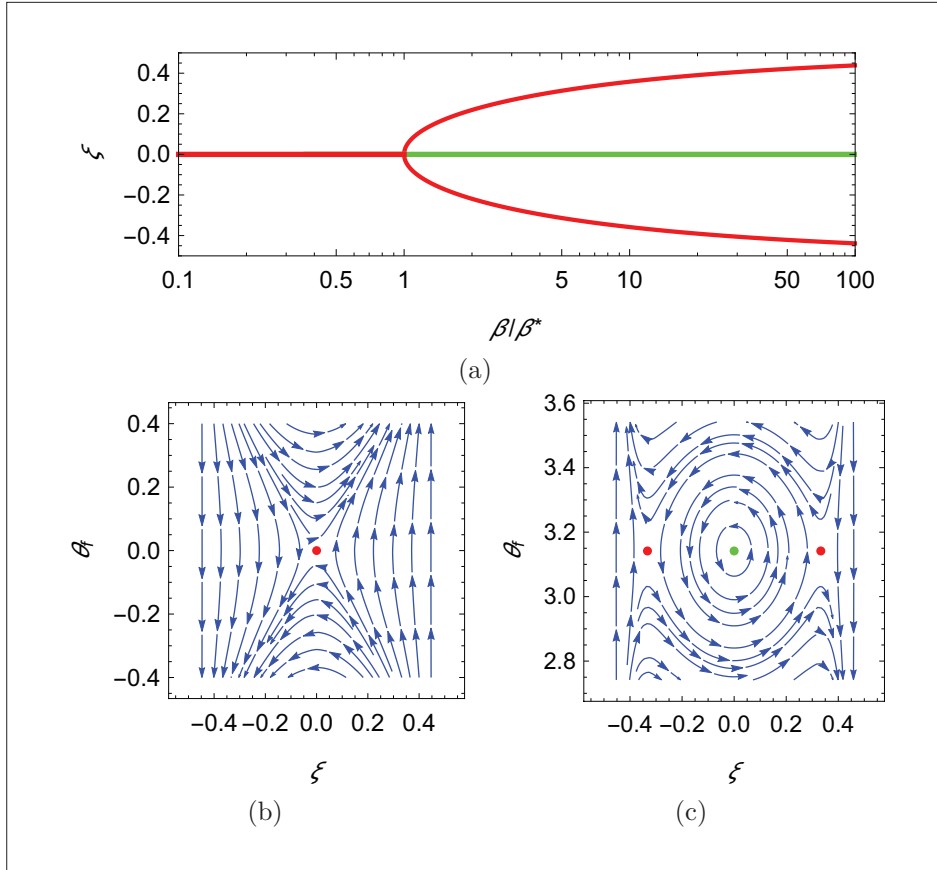

**Figure 4.** Qualitative dynamics of equation set *Equation 10a*. (**a**) Cross-stream equilibria for upstream swimming as a function of β. (**b,c**) Phase plot for downstream and upstream swimming in the case $\alpha = 0.1$, $\rho = 0.1$, and $\kappa = 1$, so that $\beta = 20$. In all panels, red refers to unstable equilibria and green to stable equilibria.

there are two solutions symmetrically located with respect to the centerline that approach the walls as $\beta \to \infty$ (*Figure 4(a)*, see Materials and methods section for mathematical derivations).

Local stability of these equilibria is determined by studying the eigenvalues of the state matrix of the corresponding linearized dynamics. For all the considered dynamics, the trace of the state matrix is zero, so that the equilibria can be saddle points (unstable) or neutral centers (stable), if the determinant is negative or positive, respectively (*Bakker, 1991*; see Materials and methods section for mathematical derivations). In the case of downstream swimming, the determinant is always negative, such that the equilibrium ($\theta_f = 0, \xi = 0$) is a saddle point (*Figure 4(b)*). For upstream swimming, the equilibrium ($\theta_f = \pi, \xi = 0$) is stable if $\beta > \beta^*$, leading to periodic oscillations similar to experimental observations (*Bak-Coleman et al., 2013*; *Bak-Coleman and Coombs, 2014*; *Elder and Coombs, 2015*; *Figure 1(b)*); the other two equilibria located away from the centerline are always unstable (*Figure 4(b and c)*). Oscillations about the centerline during rheotaxis have a radian frequency $\omega_0 \simeq (\pi^2/2)\rho\sqrt{\beta/\beta^* - 1}$, such that the frequency increases with the square root of β and is zero at $\beta^*$ (see Materials and methods section for the mathematical derivation).

## Discussion

There is overwhelming evidence that fish can negotiate complex flow environments by responding to even small flow perturbations (*Liao, 2007*). However, seldom are these perturbations included in mathematical models of fish behavior, which largely rely on vision cues (*Calovi et al., 2014*; *Couzin et al., 2005*; *Gautrais et al., 2009*; *Zienkiewicz et al., 2015*). In this paper, we proposed a hydrodynamic model for the bidirectional coupling between fish swimming and fluid flow in the absence of any sensory input but lateral line feedback – encapsulated by a simple linear feedback mechanism.

The model reduces to a nonlinear planar dynamical system for the cross-stream coordinate and orientation, of the kind that are featured in nonlinear dynamics textbooks for their elegance, analytical tractability, and broad physical interest (*Sastry, 2013*).

The planar system anticipates several of the surprising features of rheotaxis. In particular, this study provides some potential answers to the question raised by *Coombs et al., 2020*: "…what role, if any, do passive (e.g. wind vane) mechanisms play in rheotaxis and how are these influenced by fish factors (e.g. body shape) and flow dynamics?" Through the mathematical analysis of the model, we uncovered an equilibrium at the channel centerline for upstream swimming whose stability is controlled by a single non-dimensional parameter that summarizes flow speed, lateral line feedback, flow gradient, channel width, and fish size. Above a critical value of this parameter, the model predicts that rheotaxis is stable and fish will begin periodic cross-stream sweeping movements whose amplitude can be as large as the channel width. Interestingly, the model anticipates rheotaxis even without sensory feedback, through only passive hydrodynamic mechanisms.

Our mathematical proof of the existence of a nontrivial threshold for $\beta$ above which upstream swimming becomes stable finds partial support in experimental observations on a number of species in the absence of visual cues (see Appendix 3, where we have performed a bibliographical survey on experimental studies about rheotaxis). Several of these experiments have indicated the existence of a threshold in the flow speed or flow gradient above which fish successfully perform rheotaxis. Importantly, we predict that the presence of channel walls is necessary for the emergence of such a threshold, since for $\rho \to 0$, $\beta \to \infty$, thereby automatically guaranteeing the stability of upstream swimming. Based on our estimation of $\alpha$ and $\rho$ from available data, $\beta$ can be as small as $10^{-1}$ and exceed $10^2$, thereby encompassing the critical value $\beta^* \simeq 3$ (see Materials and methods section for estimation of model parameters). We should exercise care in drawing comparisons with experiments, which only control for visual feedback, in contrast with the model where we block all sensory modalities except of the lateral line. As reviewed by *Coombs et al., 2020*, water-motion cues can also be accessible to tactile or other cutaneous senses, beyond the lateral line that is included in our model. In addition, body-motion cues are not limited to visual senses, whereby they can be accessed by tactile and vestibular senses. Hence, a one-to-one comparison between experiments and theory is presently not possible.

The model predicts the emergence of rheotaxis in the absence of any sensory information. Setting $\kappa = 0$ in our model eliminates hydrodynamic feedback, yet, the fish is able to perform rheotaxis at sufficiently large flow speeds and steep flow gradients. This finding would support the possibility of a completely passive mechanism for rheotaxis. To date, there is no experiment on live animals that can be used to support this claim, owing to the necessity to eliminate all sensory modalities without compromising fish ability to swim. In practice, this may be unfeasible to do. As discussed in *Coombs et al., 2020*, existing approaches for disabling senses suffer from potential pitfalls, including: (i) unintended effects on the overall fish behavior, which are likely to occur in an effort to block at once vestibular, tactile, and lateral line senses, and (ii) difficulty in guaranteeing complete blockage of a sensory modality, which, like the lateral line, can be distributed throughout the whole body.

A potential line of approach to explore the possibility of a complete passive form of rheotaxis is through experiments with robotic fish ( *Duraisamy et al., 2019*; *Wang et al., 2020*; *Zhang et al., 2016*), mimicking locomotory patterns of live animals and allowing to precisely control sensory input. In this vein, we foresee experiments with robotic fish in a complete open-loop operation that does not utilize any sensory input. The robotic fish developed in our previous study (*Kopman and Porfiri, 2013*; *Kopman et al., 2015*) could offer a versatile platform to conduct such an experiment. Such a robot is actuated by a built-in step motor to undertake a periodic tail beating with a predetermined frequency. All its electronics is encased in the frontal section of the robot, so that its size and shape can be readily adjusted though rapid prototyping.

Although free swimming experiments on the robotic fish would be ideal, practicality may suggest to constraint the streamwise and vertical location of the robot while allowing cross-stream motions and heading changes. Such a setup shall also include a load cell to measure the drag on the robot, providing an independent measure to set the tail beat frequency – similar to CFD simulations (see Appendix 2). Also, we would recommend measuring energy expenditure by the robotic fish to gain insight into the hydrodynamic costs and benefits of rheotaxis. Experimental parameters encapsulated in $\beta$, including inlet flow speed, channel width, and robot length can be all controlled, and the flow

curvature at the centerline can be measured through velocimetry techniques. In the experiments, one shall track the motion of the robotic fish to score conditions in which stable rheotaxis is observed and extract other salient information, such as the frequency of cross-stream sweeping, if present.

The model predicts that increasing κ broadens the stable region, leading to more robust rheotaxis, which is in qualitative agreement with experimental observations – including experiments on animals with intact versus compromised lateral lines (see Appendix 3). The model prediction on the influence of the environment on rheotaxis, including the flow gradient and flow channel size, also parallels the literature on rheotaxis (see Appendix 3). For example, observations by *Oteiza et al., 2017* suggest that increasing the flow gradient $\epsilon$ enhances hydrodynamic feedback in zebrafish, resulting in improved rheotaxis.

Our analysis also indicates that wider channels should promote rheotaxis by lowering the critical speed above which swimming against the flow becomes a stable equilibrium. This mathematical finding is indirectly supported by experimental observations (see Appendix 3), and bears relevance in the design of experimental protocols for the study of rheotaxis. Confining the subject in a narrow channel will promote bidirectional hydrodynamic interactions with the walls, so that small movements of the animal will reverberate into sizeable changes in the flow physics that will mask the gradient of the background flow. Similarly, in partial alignment with experimental observations, the model predicts a lower threshold for longer fish, owing to a magnification of the hydrodynamic feedback received by a longer body (see Appendix 3). Again, we warn care in drawing comparisons due to the presence of other senses in real experiments, which are not modelled in our work.

Finally, the model anticipates the onset of periodic cross-stream sweeping, which has been studied in some experiments on fish swimming in channels without vision (*Coombs et al., 2020*). While there is not conclusive experimental evidence regarding the dependence of the frequency of oscillations on flow conditions, the model is in qualitative agreement with experiments by *Elder and Coombs, 2015*, showing a sublinear dependence on the flow speed. Therein, it is shown that the radian frequency has a weak positive tendency with respect to the flow speed for Mexican tetra swimming with or without cues from the lateral line. Above $2\,\mathrm{cm\,s^{-1}}$, the animals can successfully perform rheotaxis and display sweeping oscillations at about three cycles per minute and increase to about four cycles per minute at $12\,\mathrm{cm\,s^{-1}}$. These correspond to a radian frequency on the order of $0.1\,\mathrm{rad\,s^{-1}}$, which is similar to what we would predict for β ranging from $10^0$ to $10^1$ and $\rho$ of the order of $10^{-1}$ (recall that the time is scaled with respect to time required by the animal to traverse the channel from wall to wall in the absence of a background flow). We acknowledge that the current model does not describe contact and impact with the walls of the channel, which could be important in further detailing the onset of cross-sweeping motions that could involve stick-and-slip at the bottom of the channel (*Van Trump and McHenry, 2013*).

Just as other minimal models of fish swimming have helped resolve open questions on scaling laws (*Gazzola et al., 2014*), gait (*Gazzola et al., 2015*), and drag (*Sánchez-Rodríguez et al., 2020*), the proposed effort addresses some of the baffling aspects of rheotaxis through a transparent and intuitive treatment of bidirectional hydrodynamic interactions between fish and their surroundings. The crucial role of these bidirectional interactions hints that active manipulation of their surroundings by fish offers them a pathway to overcome sensory deprivation and sustain stable rheotaxis.

The proposed model is not free of limitations, which should be addressed in future research. The current model neglects the elasticity and inertia of the fish, which might reduce the accuracy in the prediction of rheotaxis, especially transient phenomena. Future research should refine the dipole paradigm toward a dynamic, unsteady model that accounts for added mass effects and distributed elasticity, similar to those used in the study of swimming robots (*Colgate and Lynch, 2004*; *Sfakiotakis et al., 1999*). The model could also be expanded to account for additional sensory modalities, such as vision, vestibular system, and tactile sensors on the fish body surface. We argue that pursuing any of these extensions shall require detailed experimental data, beyond what the literature can currently offer. Experiments are also needed to refine the linear hydrodynamic feedback mechanism that we hypothesized for the lateral line; in this vein, future experiments could be designed to parametrically vary the flow speed and quantify the activity level of lateral line nerve fibers through neurophysiological recordings (*Mogdans, 2019*). Beyond the inclusion and refinement of individual sensory modalities, we envision research toward the incorporation of a multisensory framework, as the one introduced by *Coombs et al., 2020*. Such a framework identifies several motorsensory integration

sites in the central nervous system that could contribute to rheotaxis, thereby calling for modeling efforts at the interface of neuroscience and fluid mechanics.

Despite its limitations, the proposed minimalistic model is successful in anticipating some of the puzzling aspects of rheotaxis and points at the possibility of attaining rheotaxis in a purely passive manner, without any sensory input. Most importantly, the model brings forward a potential methodological oversight of laboratory practice in the study of rheotaxis, caused by bidirectional hydrodynamic interactions between the swimming fish and the fluid flow. To date, there is no gold standard for the selection of the size of the swimming domain, which is ultimately chosen on the basis of practical considerations, such as facilitating behavioral scoring and creating a laminar background flow. The model demonstrates that the width of the channel has a modulatory effect on the threshold speed for rheotaxis and the cross-stream swimming frequency, which challenges the comparison of different experimental studies and confounds the precise quantification of the role of individual sensory modalities on rheotaxis. Overall, our effort warrants reconsidering the behavioral phenotype of rheotaxis, by viewing fish as an invasive sensor that modifies the encompassing flow and hydrodynamically responds to it.

## Materials and methods

### Derivation of the turn rate equation for the fish dynamics

The expression for the turn rate in *Equation 8* is obtained from the original finite-dipole model by *Tchieu et al., 2012*, in the limit of small distances between the vortices in the pair ($r_0 \to 0$).

Specifically, eqution (2.11) from *Tchieu et al., 2012*, adapted to the case of a single dipole reads

$$\dot{\theta}_f = \text{Re}\left[ \frac{(\mathcal{U}(\vec{r}_{f,r}) - i\mathcal{V}(\vec{r}_{f,r})) - (\mathcal{U}(\vec{r}_{f,l}) - i\mathcal{V}(\vec{r}_{f,l}))}{r_0} e^{i\theta_f} \right], \tag{11}$$

where subscript $l$ and $r$ refer to the left and right vortices forming the pair and $\vec{\mathcal{U}} = \mathcal{U}\hat{i} + \mathcal{V}\hat{j}$ is the advective velocity field acting on the dipole. The advective field consists of the interactions with the walls and the background flow, so that $\vec{\mathcal{U}}(\vec{r}) = \vec{u}_w(\vec{r}, \vec{r}_f, \theta_f) + \vec{u}_b(\vec{r})$; in the case of *Tchieu et al., 2012*, such a field encompasses the velocity field induced by any other dipole in the plane. Left and right vortices are defined so that $\vec{r}_{f,l} = \vec{r}_f + r_0\hat{v}_f^\perp/2$ and $\vec{r}_{f,r} = \vec{r}_f - r_0\hat{v}_f^\perp/2$, which yields $\vec{r}_{f,l} - \vec{r}_{f,r} = \hat{v}_f^\perp r_0$.

By carrying out the complex algebra in *Equation 11*, we determine

$$\dot{\theta}_f = \left( \frac{-\vec{\mathcal{U}}(\vec{r}_{f,l}) + \vec{\mathcal{U}}(\vec{r}_{f,r})}{r_0} \right) \cdot \hat{v}_f, \tag{12}$$

which supports the intuition that the dipole will turn counter-clockwise if the right vortex would experience a stronger velocity along the swimming direction. Upon linearizing the term in parenthesis in the neighborhood of $\vec{r}_f$, this expression becomes

$$\dot{\theta}_f = -\nabla\vec{\mathcal{U}}(\vec{r}_f)\hat{v}_f^\perp \cdot \hat{v}_f. \tag{13}$$

The chosen approach is consistent from the standpoint of vortex dynamics, by which each vortex in the pair advects in response to local fluid velocity. In this vein, the fish is interpreted as a bluff body, which rotates according to a difference in the drag experienced by its left and right sides. Such a difference is amplified by the pectoral fins, which enhance the effect of any left-to-right asymmetry in the surrounding fluid flow. In the literature, this description is termed T-dipole, in opposition to the so-called A-dipole that introduces two fiducial points along the direction of motion of the dipole that govern its turning (*Kanso and Cheng Hou Tsang, 2014*). Whether one representation is superior to the other in terms of accuracy is yet to be clarified; our choice of using a T-dipole is based on its theoretical consistency and intuition on the underlying flow physics. Potential avenues for resolution include detailed CFD simulations of free swimming fish or experiments with robotic fish. For completeness, in Appendix 4, we report model predictions based on the A-dipole.

### Determination of the equilibria of the planar dynamical system

By setting $\theta_f = 0$ or $\theta_f = \pi$ in equation set (*Equation 10a*, *Equation 10b*), we determine that $\xi$ should be equal to some constant, which is a root of the following transcendental equation:

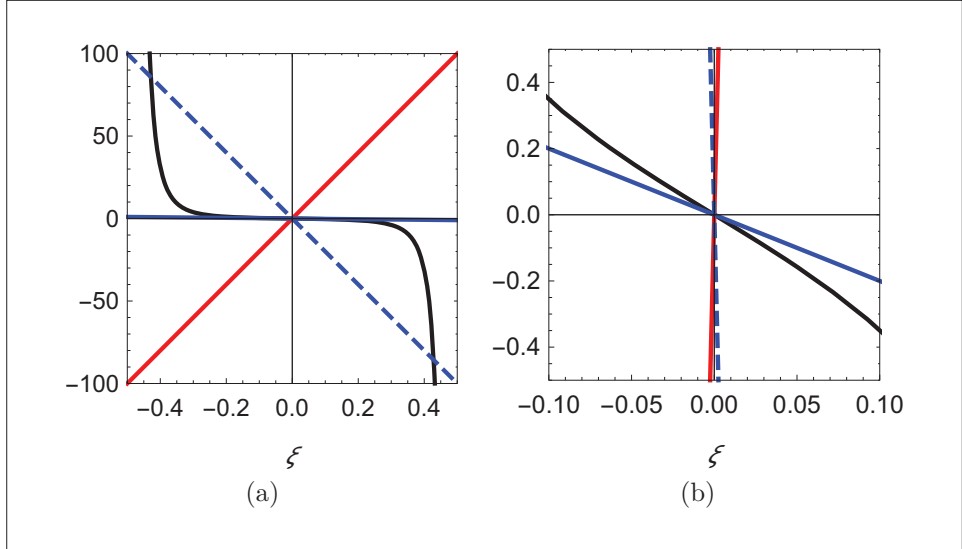

**Figure 5.** Visual illustration of the process of determining the roots of **Equation 14**. (a) Plot of the function $\frac{\pi^3}{32} \cot\left(\pi\left(\xi + \frac{1}{2}\right)\right) \csc^2\left(\pi\left(\xi + \frac{1}{2}\right)\right)$ (black), superimposed with three lines of different slope: 200 (red), -200 (dashed blue), and -2 (solid blue). (b) Zoomed-in view of the curves in (a) showing that the blue line can only intersect the black curve at the origin.

$$\frac{\pi^3}{32} \cot\left(\pi\left(\xi + \frac{1}{2}\right)\right) \csc^2\left(\pi\left(\xi + \frac{1}{2}\right)\right) = \pm\beta\xi, \tag{14}$$

where the positive sign corresponds to $\theta_f = 0$ and the negative sign to $\theta_f = \pi$. Here, $\beta = \alpha(1 + \kappa)/\rho^2$ as introduced from the main text.

As shown in **Figure 5**, for $\theta_f = 0$, there is only one root of the equation ($\xi = 0$; see the intersection between the solid red line and the solid black curve), while up to three roots can rise for $\theta_f = \pi$ depending on the value of $\beta$. For $\beta$ smaller than a critical value $\beta^*$, only $\xi = 0$ is a solution (see the intersection between the solid blue line and the solid black curve), while for $\beta > \beta^*$ two additional solutions, symmetrically located with respect to the origin emerge (see the intersections between the dashed blue line and the solid black curve). The critical value $\beta^*$ is identified by matching the slope of the black curve at $\xi = 0$, so that $\beta^* = \pi^4/32$. Notably, the two solutions symmetrically located with respect to the centerline approach the walls as $\beta \to \infty$.

## Local stability analysis of the planar dynamical system

To examine the local stability of the equilibria of the planar dynamical system, we linearize equation set (**Equation 10a**, **Equation 10b**). The state matrix of the linearized dynamics, $A$, describes the local behavior of the nonlinear system when perturbed in the vicinity of the equilibrium, that is,

$$\dot{\delta\mathbf{q}}(t) = A\,\delta\mathbf{q}(t), \tag{15}$$

where $\delta\mathbf{q} = [\delta\xi, \delta\theta_f]^{\mathrm{T}}$ is the variation about the equilibrium. The eigenvalues of the $A$ are indicative of local stability about each equilibrium.

For $\theta_f = 0$ and $\xi = 0$, the state matrix is given by

$$A = \begin{bmatrix} 0 & 1 - \frac{\pi^2\rho^2}{6} \\ 8(1+\kappa)\alpha + \frac{\pi^4\rho^2}{4} & 0 \end{bmatrix}. \tag{16}$$

Given that the trace of the matrix is zero ($\operatorname{tr} A = 0$), the analysis of the stability of the equilibrium resorts to ensuring the sign of the determinant to be positive ($\det A > 0$). Specifically, if the determinant is positive, the eigenvalues are imaginary and the equilibrium is a neutral center (stable, although not asymptotically stable), otherwise one of the eigenvalues is positive and the equilibrium is a saddle point (unstable) (**Bakker, 1991**). Hence, stability requires that

$$\tfrac{1}{24}\left(-6 + \pi^2\rho^2\right)\left(32\alpha(1 + \kappa) + \pi^4\rho^2\right) > 0. \tag{17}$$

Since the first factor is always negative ($\rho \ll 1$) and the second is positive, the inequality is never fulfilled and the equilibrium is a saddle point (unstable) (***Figure 4(a and b)***).

For $\theta_f = \pi$ and $\xi = 0$, the state matrix is given by

$$A = \begin{bmatrix} 0 & -1 + \frac{\pi^2\rho^2}{6} \\ 8(1 + \kappa)\alpha - \frac{\pi^4\rho^2}{4} & 0 \end{bmatrix}. \tag{18}$$

Similar to the previous case, stability requires that $\det A > 0$, that is,

$$\tfrac{1}{24}\left(-6 + \pi^2\rho^2\right)\left(-32\alpha(1 + \kappa) + \pi^4\rho^2\right) > 0. \tag{19}$$

Due to the sign change in the first summand appearing in the second factor with respect to the previous case, stability becomes possible. Specifically, the equilibrium is a neutral center (stable) for $\beta > \beta^* = \pi^4/32$, which is also the necessary condition for the existence of the two equilibria symmetrically located with respect to the channel centerline (***Figure 4(a and c)***).

When $\beta > \beta^*$, we register the presence of two more equilibria at $\pm\xi \neq 0$. The state matrix takes the form

$$A = \begin{bmatrix} 0 & -1 - \frac{\pi^2\rho^2}{12} + \frac{1}{4}\pi^2\rho^2 \sec^2(\pi\xi) \\ 8(1 + \kappa)\alpha - \frac{1}{4}\pi^4\rho^2(2 - \cos(2\pi\xi))\sec^4(\pi\xi) & 0 \end{bmatrix}, \tag{20}$$

Also in this case, stability requires that $\det A > 0$, that is,

$$\tfrac{1}{48}\left(-12 + 3\pi^2\rho^2 \sec^2(\pi\xi) - \pi^2\rho^2\right)\left(-32\alpha(1 + \kappa) + \pi^4\rho^2(2 - \cos(2\pi\xi))\sec^4(\pi\xi)\right) > 0 \tag{21}$$

Once again, for $\rho \ll 1$, we can assume that the first factor in parenthesis is negative. (This assumption is grounded upon ***Equation 14***, which yields that $(\xi \pm 1/2) = \mathcal{O}(\rho^{2/3})$; since $\cos(\pi\xi)^2 = \mathcal{O}((\xi \pm 1/2)^2)$, we have that $\rho^2 \sec^2(\pi\xi) \to 0$ as $\rho \to 0$.) Hence, we obtain

$$\beta > \tfrac{\pi^4}{32}(2 - \cos(2\pi\xi))\sec^4(\pi\xi), \tag{22}$$

which is not satisfied for any choice of $\beta > \beta^*$. Thus, the two equilibria away from the channel centerline, close to the walls are always saddle points (unstable) (***Figure 4(a and c)***).

We comment that the local stability analysis requires only knowledge of the curvature of the flow field at the centerline of the channel. Hence, should one contemplate alternative profiles for the background flow, linear stability results shall not change. Higher-order parameterizations for the flow profile will result into nonlinear dependencies on $\xi$ that do not affect the linear analysis. Likewise, while we considered a linear feedback mechanism to integrate lateral information via a simple gain, one may explore nonlinear relationships between $\lambda$ and $\Gamma$. The linear stability analysis shall not change, whereby these nonlinear forms will result into dependencies on higher powers of $\xi$.

## Frequency of cross-stream sweeping

The linearized planar system about the stable focus in ***Equation 18*** is equivalent to a classical second-order system in terms of the cross-stream coordinate, similar to a mass-spring model. Hence, the radian resonance frequency of the system is

$$\omega_0 = \sqrt{\det A} \simeq \tfrac{\pi^2}{2}\rho\sqrt{\tfrac{\beta}{\beta^*} - 1}. \tag{23}$$

where the last approximation holds for $\rho \ll 1$. ***Equation (23)*** shows that, close to the threshold, the frequency of oscillations is small and it increases with β and $\rho$.

## Estimation of model parameters

In a typical experimental setup on rheotaxis, the width of the channel, $h$, is on the order of three to ten times the body length of the animal, $l$. For example, experiments from **Elder and Coombs, 2015** on Mexican tetras of $l = 8.3\,\text{cm}$ were conducted in a channel with $h = 25\,\text{cm}$. Similarly, in the experiments on adult zebrafish from **Burbano-L and Porfiri, 2021**, $l = 3.6\,\text{cm}$ and $h = 13.8\,\text{cm}$, and in the experiments on zebrafish larvae from **Oteiza et al., 2017**, $l = 4.2\,\text{mm}$ (inferred from the animals' age) and $h = 1.27 - 4.76\,\text{cm}$. The distance between the vortices simulating a fish, $r_0$, should be on the order of a tail beat, which has a typical value of $0.2l$ (**Gazzola et al., 2014**). As a result, it is tenable to assume that $\rho^2$ is between $10^{-4}$ and $10^{-2}$.

A safe estimation of the velocity of the animal in the absence of the background flow, $v_0$, would be on the order of few body lengths per second (**Gazzola et al., 2014**). The speed used for the background flow across experiments, $U_0$, tend to be of the same order as the magnitude of $v_0$, leaning toward values close to one body length per second (**Coombs et al., 2020**). For instance, data on zebrafish from **Burbano-L and Porfiri, 2021** suggest $v_0 = 5.7\,\text{cm s}^{-1}$ and $U_0 = 3.2\,\text{cm s}^{-1}$. The estimation of the non-dimensional parameter $\epsilon$ associated with the shear in the flow is more difficult, since data on the velocity profiles are seldom reported. That being said, for channel flow of sufficiently high Reynolds number, the velocity profile in the channel is expected to be blunt, approximating a uniform flow profile near the channel center (**White, 1974**). Thus, it is tenable to treat $\epsilon$ as a small parameter, between $10^{-2}$ and $10^{-1}$. For flow of low Reynolds number (**Oteiza et al., 2017**) ($\text{Re} < 100$), the velocity gradient in the channel has been observed to be large, corresponding to $\epsilon$ values in the range of $10^{-1}$ and 1. By combining these estimations, we propose that $\alpha$ ranges between 0 and 1.

An estimation of $\kappa$ is difficult to offer, whereby feedback from the lateral line has only been included in few studies (**Burbano-L and Porfiri, 2021**; **Chicoli et al., 2015**; **Colvert and Kanso, 2016**; **Oteiza et al., 2017**). Using the data-driven model from **Burbano-L and Porfiri, 2021**, it is tenable to assume values on the order of $10^1$ for individuals showing high rheotactic performance. This gain can also be estimated by comparing the threshold speeds of fish, $U_c$, with and without the lateral line, through $\frac{U_c(\text{LL}-)}{U_c(\text{LL}+)} = 1 + \kappa$, according to **Equation 29** in Appendix 3. The significant increase in the threshold speed following lateral line ablation in **Baker and Montgomery, 1999** indicates that $\kappa \in [2, 7]$, while the indistinguishable threshold speed between LL+ and LL- fish in a few other studies (**Bak-Coleman and Coombs, 2014**; **Elder and Coombs, 2015**; **Van Trump and McHenry, 2013**) may suggest that

**Table 1.** Estimation of model parameters from data in the literature.

| Reference | $\rho$ | $\epsilon$ | $\alpha$ | $\kappa$ | $\beta$ |
|---|---|---|---|---|---|
| *Bak-Coleman et al., 2013* | $\sim 0.05$ | $[10^{-2}, 10^{-1}]$ | $[0, 0.17]$ | — | — |
| *Bak-Coleman and Coombs, 2014* | $\sim 0.04$ | $[10^{-2}, 10^{-1}]$ | $[0, 0.16]$ | $\sim 0$ | $[0, 100]$ |
| *Baker and Montgomery, 1999* and *Montgomery et al., 1997* | $\sim 0.1$ | $[10^{-2}, 10^{-1}]$ | *$[0, 0.32]$ | $[2, 7]$ | $[0, 256]$ |
| *Elder and Coombs, 2015* | $\sim 0.066$ | $[10^{-2}, 10^{-1}]$ | *$[0, 0.24]$ | $\sim 0$ | $[0, 55]$ |
| *Kulpa et al., 2015* | $\sim 0.04$ | $\sim 1$ near center of jet | $\sim 1.3$ | — | — |
| *Oteiza et al., 2017* | $[0.018, 0.066]$ | $[0.20, 0.82]$ | — | — | — |
| *Peimani et al., 2017* | $\sim 0.044$ | $\sim 1$ | — | — | — |
| *Suli et al., 2012* | $\sim 0.018$ | $[0.1, 1]$ | — | — | — |
| *Van Trump and McHenry, 2013* | $[0.055, 0.127]$ | $[10^{-2}, 10^{-1}]$ | *$[0, 0.32]$ | $\sim 0$ | $[0, 106]$ |

LL+ cavefish swimming speed $v_0 \sim 5\,\text{cm/s}$ in zero background flow in **Bak-Coleman and Coombs, 2014** is used to estimate α.

$\kappa \sim 0$. In *Table 1*, we summarize the model parameters identified from data in the experimental studies detailed in Appendix 3.

## Acknowledgements

The authors acknowledge financial support from the National Science Foundation under Grant No. CMMI-1901697. The authors acknowledge Alain Boldini and Simone Macrì for useful discussions.

## Additional information

### Funding

| Funder | Grant reference number | Author |
|---|---|---|
| National Science Foundation | CMMI-1901697 | Maurizio Porfiri |

The funders had no role in study design, data collection and interpretation, or the decision to submit the work for publication.

### Author contributions

Maurizio Porfiri, Conceptualization, Data curation, Formal analysis, Funding acquisition, Investigation, M.P. conceived the study, developed the theoretical model and performed data analysis, wrote a first draft of the manuscript, which was consolidated in its present form by all the authors., Methodology, Project administration, Resources, Software, Supervision, Validation, Visualization, Writing – original draft, Writing – review and editing; Peng Zhang, Data curation, Investigation, Methodology, P.Z. conducted the literature review and performed the model validation. The manuscript was consolidated in its present form by all the authors., Validation, Writing – original draft; Sean D Peterson, Conceptualization, Data curation, Formal analysis, Funding acquisition, Investigation, Methodology, Project administration, Resources, S.D.P. conceived the study, developed the theoretical model and performed data analysis, wrote a first draft of the manuscript, which was consolidated in its present form by all the authors., Software, Supervision, Validation, Visualization, Writing – original draft, Writing – review and editing

### Author ORCIDs

Maurizio Porfiri ⬤ http://orcid.org/0000-0002-1480-3539
Peng Zhang ⬤ http://orcid.org/0000-0001-8237-1259
Sean D Peterson ⬤ http://orcid.org/0000-0001-8746-2491

### Decision letter and Author response

Decision letter https://doi.org/10.7554/eLife.75225.sa1
Author response https://doi.org/10.7554/eLife.75225.sa2

## Additional files

### Supplementary files
• Transparent reporting form

### Data availability

The authors declare that the data supporting the findings of this study are available within the paper. The Mathematica notebook used to derive the governing equations, study the planar dynamical, and generate associated figures, together with the CFD data discussed in the text, are also available at https://github.com/dynamicalsystemslaboratory/Rheotaxis; (copy archived at swh:1:rev:0c2828a439cb0df9971cdf12dc7dd93d16ce3450).

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

# Appendix 1

## Complete expression for the velocity field caused by image dipoles

The velocity field $\vec{u}_f = u_f \hat{i} + v_f \hat{j}$ at $\vec{r}$ induced by the single dipole at $\vec{r}_f$, given by the potential function in *Equation 1*, is

$$u_f(\vec{r}, \vec{r}_f, \theta_f) = r_0^2 v_0 \left( \frac{((x-x_f)^2 - (y-y_f)^2) \cos\theta_f + 2(x-x_f)(y-y_f) \sin\theta_f}{((x-x_f)^2 + (y-y_f)^2)^2} \right), \tag{24a}$$

$$v_f(\vec{r}, \vec{r}_f, \theta_f) = r_0^2 v_0 \left( \frac{-((x-x_f)^2 - (y-y_f)^2) \sin\theta_f + 2(x-x_f)(y-y_f) \cos\theta_f}{((x-x_f)^2 + (y-y_f)^2)^2} \right). \tag{24b}$$

The potential function describing the image vortex system for a dipole in a channel presented in *Equation 3* can be simplified using `Mathematica`, yielding

$$\phi_w(\vec{r}, \vec{r}_f, \theta_f) = \frac{r_0^2 v_0}{4} \left[ 4 \frac{(x-x_f)\cos\theta_f + (y-y_f)\sin\theta_f}{(x-x_f)^2 + (y-y_f)^2} \right.$$
$$\left. - \frac{\pi e^{-i\theta_f}}{h} \left( e^{2i\theta_f} \left( \coth(\pi A) + \coth(\pi B^*) \right) + \coth(\pi A^*) + \coth(\pi B) \right) \right], \tag{25}$$

where $A = ((x - x_f) + i(y - y_f))/(2h)$, $B = ((x - x_f) + i(y + y_f))/(2h)$, $i = \sqrt{-1}$, and a superscript $*$ indicates complex conjugate. The velocity field at due to the walls, is

$$v_w = \frac{r_0^2 v_0}{4} \left[ \frac{\pi^2 e^{-i\theta_f}}{2h^2} \left( e^{2i\theta_f}(\operatorname{csch}^2 \pi A + \operatorname{csch}^2 \pi B^*) + (\operatorname{csch}^2 \pi A^* + \operatorname{csch}^2 \pi B) \right) \right.$$
$$\left. + \frac{4\cos\theta_f}{(x-x_f)^2 + (y-y_f)^2} - \frac{8(x-x_f)((x-x_f)\cos\theta_f + (y-y_f)\sin\theta_f)}{((x-x_f)^2 + (y-y_f)^2)^2} \right]. \tag{26a}$$

$$v_w = \frac{r_0^2 v_0}{4} \left[ \frac{i\pi^2 e^{-i\theta_f}}{2h^2} \left( e^{2i\theta_f}(\operatorname{csch}^2 \pi A - \operatorname{csch}^2 \pi B^*) - (\operatorname{csch}^2 \pi A^* + \operatorname{csch}^2 \pi B) \right) \right.$$
$$\left. + \frac{4\sin\theta_f}{(x-x_f)^2 + (y-y_f)^2} - \frac{8(y-y_f)((x-x_f)\cos\theta_f + (y-y_f)\sin\theta_f)}{((x-x_f)^2 + (y-y_f)^2)^2} \right]. \tag{26b}$$

Superimposing the velocity fields from the dipole and its images and setting $y = 0$ (or $y = h$) yields $v_f + v_w = 0$, thereby confirming that the walls of the channel are streamlines.

## Appendix 2

## Computational fluid dynamics

### Framework

To quantify the flow around the swimming fish, we solved the incompressible Navier-Stokes equations numerically in the commercial software COMSOL Multiphysics (version 5.6). We focused on the steady swimming of a giant danio exhibiting a carangiform swimming pattern, consisting of large body undulations in the posterior of the animal and minimal lateral movement in the anterior portion. The lateral movement of a giant danio was mathematically described through a local coordinate system, $x'$-$y'$, such that the undeformed fish aligns its centerline along the $x'$-axis with head at $x' = 0$ and tail at $x' = l$ (see **Appendix 2—figure 1a**). The lateral movement were described as a traveling wave from the head to the tail as (**Najafi and Abtahi, 2022**).

$$y' = \left[ c_1 \left( x' - 0.3l \right) + c_2 \left( x' - 0.3l \right)^2 \right] \sin \left[ k_L \left( x' - 0.3l \right) - 2\pi ft \right] \tag{27}$$

where $c_1$ and $c_2$ are two constants that describe the shape of the undulation envelope, $k_L$ is the wavenumber, and $f$ is the tail beating frequency. The values of $l$, $c_1$, $c_2$, and $k_L$ are taken from **Najafi and Abtahi, 2022**, and the value of $f$ is taken from **Zhang et al., 2019**; these parameters are summarized in **Appendix 2—table 1**. For the chosen tail beat frequency, the oscillatory Reynolds number (**Gazzola et al., 2014**) is $\mathrm{Sw} = \frac{2\pi Cfl}{\nu} = 8,100$, where $C$ is the tail beat amplitude, given as $C = 0.7 \times c_1 l + 0.49 \times c_2 l^2$, and $\nu$ is the kinematic viscosity of water at room temperature ($\nu = 1.0 \times 10^{-6}\,\mathrm{m^2 s^{-1}}$).

The undeformed giant danio body shape was approximated as a NACA0013 airfoil, which has a maximum thickness-to-chord length matching that of the animal in **Zhang et al., 2019**. A NACA airfoil offers a reasonable approximation of the fish body in terms of its streamlined shape and its ability to emulate thrust production of a swimming fish (**Lucas et al., 2020**). The movement of the fish body was imposed in the numerical simulations using a moving boundary. No-slip boundary conditions was set on the fish body, whereas the channel walls satisfied only the no-penetration boundary condition, that is, they were treated as slip walls. In the simulations, we set a uniform flow with speed $U_0$ at the inlet, imposed a zero pressure boundary condition at the outlet, and fixed the axial location of the fish at the channel centerline.

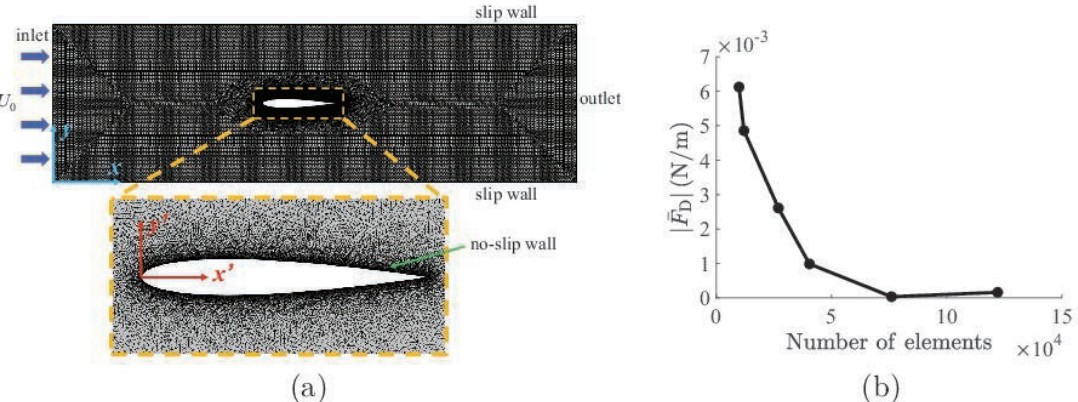

**Appendix 2—figure 1.** Details about the implementation of the computational fluid dynamics simulations. (**a**) Mesh implemented in the simulations, with definitions of coordinate systems and a zoomed-in view of the refined mesh around the fish. (**b**) Mesh convergence analysis, showing the mean drag as a function of number of elements in the simulation.

The fluid domain was discretized using triangular elements, as illustrated in **Appendix 2—figure 1a**, and was allowed to deform in time to accommodate the body undulations. A grid convergence study was performed to ensure the solution was independent of the mesh. As shown in **Appendix 2—figure 2b**, a total number of $76\,\mathrm{k}$ elements were sufficient to guarantee the mesh independence of the simulations, such that implementing a higher number of $122\,\mathrm{k}$ elements introduced negligible variation in the mean drag prediction on the fish. As a result, all simulations were conducted

using $76\,\mathrm{k}$ elements. A total of 10 tail beating cycles were simulated in each simulation, which was sufficiently long for the flow in the channel to fully develop. All simulations were conducted on 12 Intel Xeon Platinum 8268 CPUs with a base frequency of $2.90\,\mathrm{GHz}$ and a total of $48\,\mathrm{GB}$ memory. A typical simulation required approximately $6\,\mathrm{h}$ computational time to complete.

To represent the realistic swimming condition, the imposed inlet flow speed, $U_0$, should match the fish swimming speed, $v_0$, which depends on the body undulations described by (27). To identify the value of $v_0$, we conducted a series of preliminary simulations by varying $U_0$ until the time-averaged total drag on the fish, $\bar{F}_\mathrm{D}$ was zero. That is, the drag experienced by the fish exactly balanced the thrust generated by the body undulations. The resulting swimming speed was determined to be $v_0 = 19.74\,\mathrm{cm/s}$. The resulting Reynolds number based upon channel width is $\mathrm{Re} = \frac{hU_0}{\nu} = 29,600$.

## Results

The instantaneous velocity fields in the vicinity of the giant danio relative to the background flow are presented at five time instants during half of a tail beating cycle in *Appendix 2—figure 2*. In comparison with the mean flow shown in *Figure 3*, we observe distortions of the streamlines, with a left-right asymmetry caused by the body undulations. We also identify a series of alternating vortices generated through tail beating that form the wake. The magnitude of the wake flow decays as it is advected downstream.

CFD results are also useful to quantify the thickness of the boundary layer along the animal and offer support in favor of the proposed inviscid model for the study of the interactions between a fish and the channel walls. The thickness of the boundary layer can be quantified from the flow velocity component along the $x$-direction, $u_\mathrm{t}$, in the presence of the background flow. As shown in *Appendix 2—figure 3*, the value of $u_\mathrm{t}$ is zero at the body surface due to the no-slip boundary condition. A large gradient is observed within a thin layer at the boundary, in which $u_\mathrm{t}$ rapidly increases to $v_0$. The boundary layer thickness can be estimated by identifying the location at which $u_\mathrm{t}$ reaches 99% of its asymptotic value away from the fish body. During a tail beating cycle, the boundary layer thickness ranged between 2.8% and 15.7% of the fish body length, thereby supporting the feasibility of neglecting the influence of the viscous boundary layers as a first-order approximation.

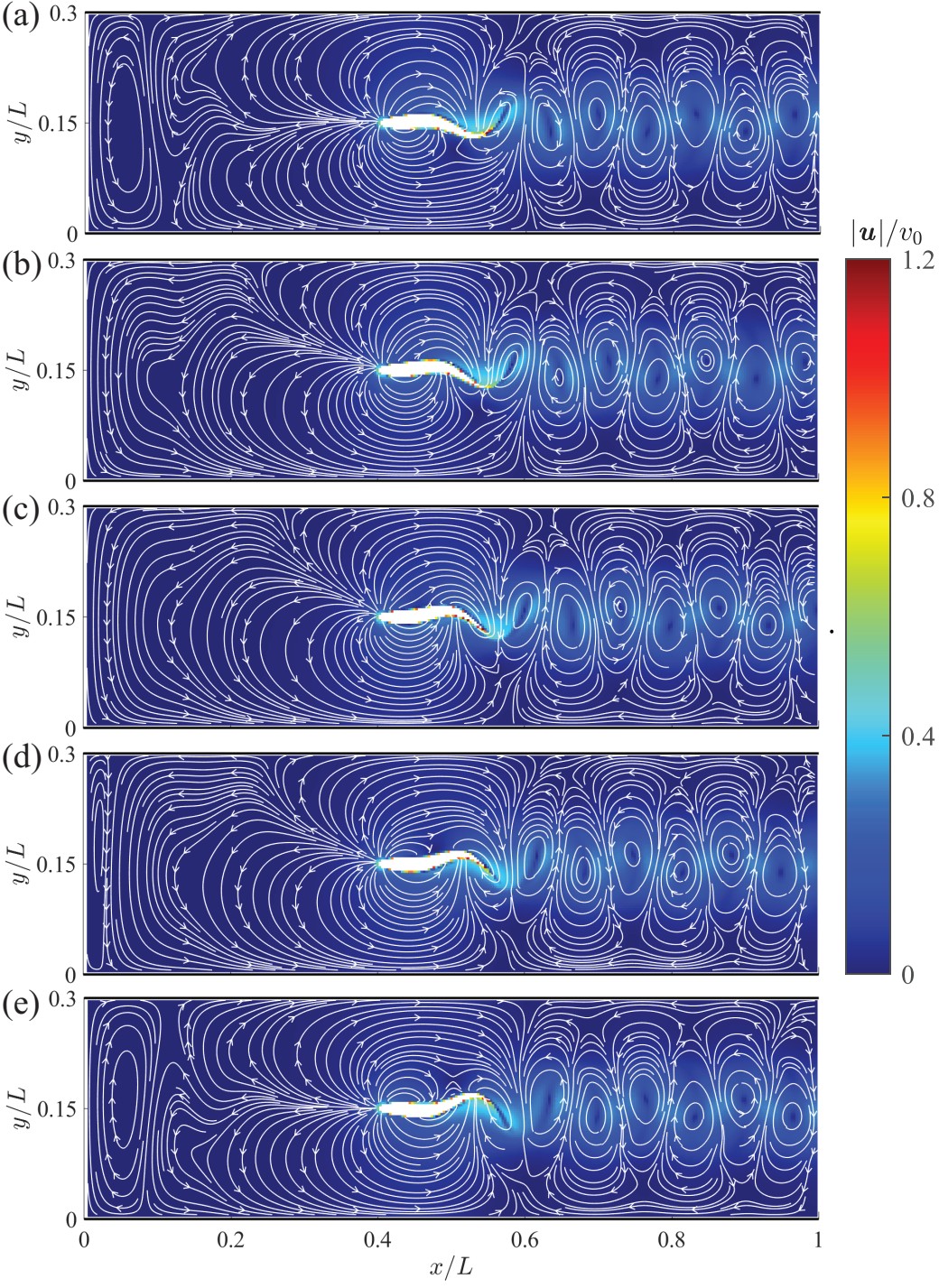

**Appendix 2—figure 2.** Instantaneous velocity fields around a swimming fish relative to the background flow from computational fluid dynamics. (**a**) – (**e**) correspond to $t = 0$, $T/8$, $T/4$, $3T/8$, and $T/2$, respectively, where $T$ is the period of a tail beat. White curves are streamlines with arrows indicating flow directions.

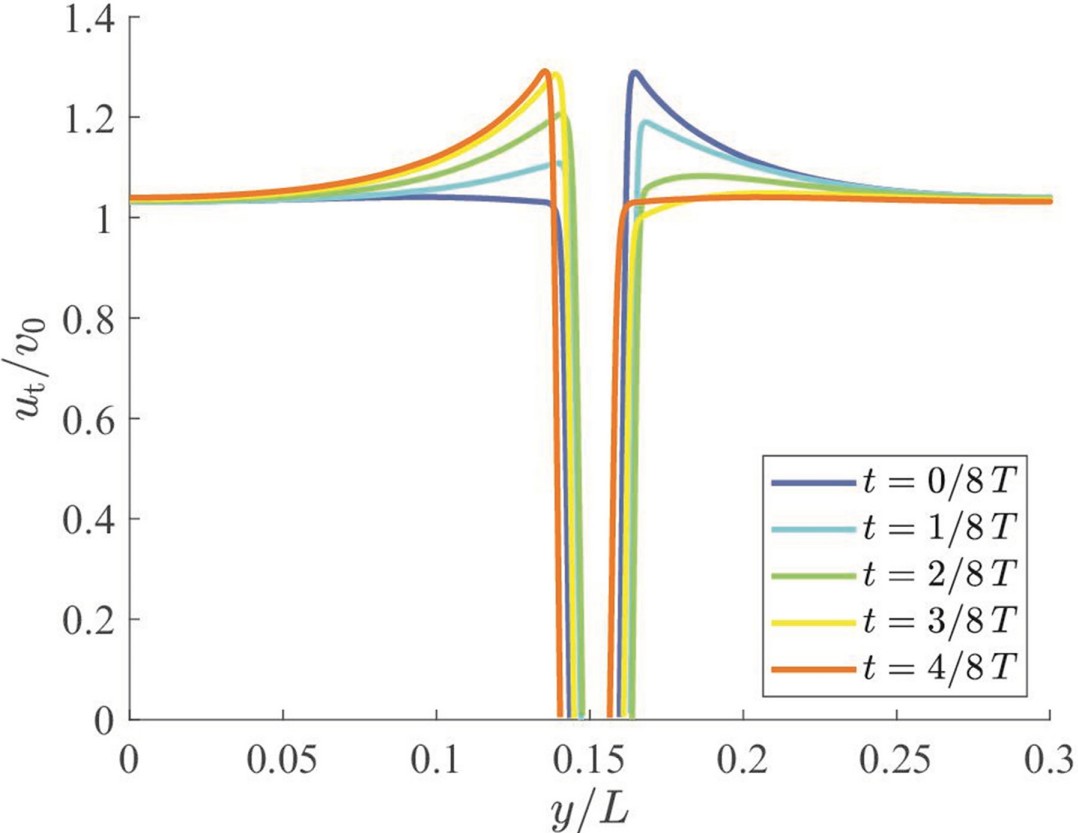

**Appendix 2—figure 3.** Analysis of the boundary layer thickness along a swimming fish. x-component of the flow velocity, $u_t$, extracted across the half-length of the fish body. The values of $u_t$ are measured in a coordinate system moving at the speed of $v_0$ along the fish swimming direction.

## Comparison with the dipole model

The velocity field predicted by the dipole model is validated against the mean velocity quantified through CFD (see *Figure 3(a)*). Consistent with the fish location and heading direction, we set $y_f = h/2$ and $\theta = \pi$ for the dipole. The axial location, $x_f$, and the characteristic length scale, $r_0$, of the dipole are treated as fitting parameters. The velocity field associated with the dipole model in *Figure 3(b)* corresponds to a set of fitting parameters that minimizes the discrepancy between the model prediction and the simulated simulation. Denoting the model-predicted velocity field as $\vec{u}^{\text{dipole}}$ and the simulated one as $\vec{u}^{\text{CFD}}$, the discrepancy between them is quantified through

$$E = \sqrt{\frac{\int_{\mathcal{D}} \left| \vec{u}^{\text{dipole}} - \vec{u}^{\text{CFD}} \right|^2 \, \mathrm{d}A}{\int_{\mathcal{D}} \, \mathrm{d}A}}, \tag{28}$$

where $\mathcal{D}$ is the computational domain. The optimal values of the fitting parameters that minimize $E$ are determined through an exhaustive search of the parameter space, leading to $(x_f - x_s) = 0.315l$ and $r_0 = 0.062l$, where $x_s$ is the axial location of the fish head (the origin of the $x'$-$y'$ coordinate system).

**Appendix 2—table 1.** Parameters employed in (27) to describe the locomotory pattern of a giant danio.

| Parameters | $l$ | $c_1$ | $c_2$ | $k_L$ | $f$ |
|---|---|---|---|---|---|
| Values | 7.3 cm | 0.004 | $-2.33\,\mathrm{m}^{-1}$ | $78.5\,\mathrm{m}^{-1}$ | 3 Hz |

# Appendix 3

## Bibliographical survey

We surveyed over three hundred publications cited by *Arnold, 1974* and *Coombs et al., 2020* – two review papers on rheotaxis, with the former focusing on early investigations from 1900 s to 1970 s, and the latter highlighting more recent works conducted between 1970 s and 2020. Publications were selected through the following inclusion and exclusion criteria.

## Criteria

### Inclusion criteria

We selected studies where: (i) the subject animals were fish; (ii) fish demonstrated rheotactic behavior; (iii) no unsteady flow events were present in the swimming domain, such as the wake structure of obstacles; (iv) the sensory cues available to fish could be identified with some confidence; (v) fish behavior was not influenced by social interactions; (vi) fish swam without visual cues; and (vii) the publication was written in English. Within criterion (iii), we focused on experiments with steady flows where the flow gradient is consistent over time, thus excluding swimming in random flow events. Criterion (vi) was introduced to direct our search toward the effects of hydrodynamic cues and lateral line sensing, which limited our search to experiments using blind fish or experiments in the dark. We acknowledge that this condition is not reflective of the model hypotheses, which, in fact, block any sense except for the lateral lines (such as vestibular and tactile senses).

### Exclusion criteria

Among studies identified through the selection criteria, we excluded experiments on pleuronectiform flatfishes, which swim on their side and generate propulsive undulations in a vertical plane (*Webb, 2002*). This locomotory pattern differs fundamentally from the current model, derived on the assumption the fish align their bodies vertically and undulate on a horizontal plane, which is the swimming strategy of the majority of fishes.

## Dataset

*Appendix 3—table 1* presents data extracted from the selected studies, including the fish species, size of the swimming domain, flow conditions, sensory cues available to the fish, and the measured rheotaxis threshold speed. Swimming domains with rectangular cross-sections are defined by their length ($L$), width ($h$), and depth ($W$), while swimming domains with circular cross-sections by their length and diameter ($D$). Flow conditions are quantified through the flow speed and flow gradient. If information about the flow gradient was not available in a study, we qualitatively estimated its value through the Reynolds number of the flow, defined based on the width (diameter) of the channel and the background flow speed as $\mathrm{Re} = \frac{hU_0}{\nu}$ ($\mathrm{Re} = \frac{DU_0}{\nu}$). For a sufficiently high $\mathrm{Re}$, the flow gradient near the center of the channel is expected to be low.

## Comparison against model predictions

The studies identified in *Appendix 3—table 1* are utilized to offer some context to the proposed theoretical framework with respect to the rheotaxis stability threshold. We first express the stability threshold $\beta = \beta^*$ in dimensional form in terms of the rheotaxis threshold speed

$$U_c = \frac{\pi^4 r_0^2 v_0}{32 h^2 (1 + K r_0 l) \epsilon}, \tag{29}$$

such that $U_0 > U_c$ corresponds to the stable condition $\beta > \beta^*$, and vice versa. From most studies, the values of $U_c$ can be identified and its confidence level can be inferred (see *Appendix 3—table 1*).

As evidenced in *Equation 29*, a series of parameters could influence the rheotaxis threshold speed, including the lateral line feedback, flow gradient, swimming domain size, and fish body length. Specifically, *Equation 29* predicts that increasing the lateral line feedback, flow gradient, and/or width of swimming channel promotes rheotaxis at lower flow speeds, whereas increasing fish size will require higher flow speeds to elicit rheotactic behavior. The effects of these parameters are validated independently in *Appendix 3—table 2*, where we include experimental evidence garnered within each study and, when possible, carry out a comparison, across them. We compare each of these empirical observations to model predictions, and assess if they support the model, contradict the model, or are inconclusive. An observation is considered supportive of (contradictive to) our

model if the measured $U_c$ exhibits with statistical significance the same (opposite) dependence on a certain parameter. Data that lack statistical significance are considered inconclusive.

The confidence intervals of the measured $U_c$ values were estimated to determine if $U_c$ were significantly different across studies. When the mean and standard error of the mean (s.e.m.) of $U_c$ were provided, we estimated its 95% confidence interval as (mean $-$ 1.96 s.e.m., mean $+$ 1.96 s.e.m.). If the confidence intervals of two $U_c$ values did not overlap, we considered them significantly different. For instance, in **Bak-Coleman and Coombs, 2014** and **Elder and Coombs, 2015**, the confidence intervals of $U_c$ were determined to be (0.63, 1.17) and (1.27, 2.64) cm/s, respectively, and thus the $U_c$ values were considered significantly different. In several studies, such as **Baker and Montgomery, 1999** and **Van Trump and McHenry, 2013**, the threshold speeds were only estimated as intervals, where fish swimming below a lower bound did not perform rheotaxis, while they exhibited rheotaxis above an upper bound. We treated this speed interval as the confidence interval for $U_c$ in our statistical analysis.

## Effect of lateral line feedback

Several studies provide some support in favor of the prediction of our model of the beneficial role of lateral line feedback, showing a significant reduction in rheotactic performance when the lateral line is compromised (**Kulpa et al., 2015**; **Oteiza et al., 2017**; **Suli et al., 2012**), see **Appendix 3—table 2**. In these studies, fish locomotion was measured in steady background flows, so that a fish holding station would experience minimal linear acceleration and marginally engage the vestibular system. Throughout these studies, fish were not observed to make contact with the swim channel, indicating that tactile senses played a negligible role in rheotaxis.

## Effect of flow gradient

We identified two studies (**Lyon, 1904**; **Oteiza et al., 2017**) that could back the predicted effect of the flow velocity gradient on rheotaxis, as summarized in **Appendix 3—table 2**. In both studies, fish locomotion was recorded in flows with varying velocity gradients. In qualitative agreement with the proposed model, the rheotaxis performance of zebrafish larvae significantly improved with increasing gradient magnitudes (**Oteiza et al., 2017**). Similar observations were obtained by Lyon on blind Fundulus (**Lyon, 1904**), where in a flow with a small gradient, fish performed rheotaxis only when tactile cues were available, while in a jet flow with a large flow gradient, rheotaxis could be elicited solely by the flow. Although qualitatively in line with our predictions, we conservatively considered this study as inconclusive due to a lack of quantitative data for statistical tests.

## Effect of size of swimming domain

To elucidate the role of the swimming domain size on rheotaxis threshold speed, we conducted cross-study comparisons as shown in **Appendix 3—table 2**. As evidenced through comparisons between two experiments on zebrafish larvae (**Oteiza et al., 2017**; **Peimani et al., 2017**) in swim channels of drastically different sizes, rheotaxis was elicited at a higher threshold speed in a smaller flow channel, which supports our model prediction. Our model is also qualitatively supported by comparisons between **Bak-Coleman and Coombs, 2014** and **Baker and Montgomery, 1999**, or **Bak-Coleman and Coombs, 2014** and **Van Trump and McHenry, 2013**, where experiments on blind cavefish of comparable body sizes uncovered higher threshold speeds in smaller flow channels. In the experiments of **Bak-Coleman and Coombs, 2014**, blind cavefish were observed to receive transient tactile senses while swimming, which could have contributed to its lower rheotaxis threshold. As a result, experimental data on blind cavefish were conservatively deemed to be inconclusive.

## Effect of fish size

The relationship between the threshold speed and fish body size is less straightforward, as the body size not only determines the value of $l$, but also influences $r_0$, which is on the order of the fish tail beat amplitude. We assumed $r_0 = 0.2l$, which is a typical tail-beat-amplitude-to-body-length ratio (**Gazzola et al., 2014**). For fish with functional lateral lines that produce positive feedback, $K > 0$, we obtain $U_c \sim 1 - \frac{1}{1+0.2Kl^2}$. For fish with disabled lateral line, $K = 0$, we find $U_c \sim l^2$. In both cases, the model predicts that $U_c$ is larger for fish with larger body length. Some evidence can be garnered by contrasting a pair of studies by **Bak-Coleman and Coombs, 2014** and **Elder and Coombs, 2015**, where experiments were conducted on fish of the same species (*Astyanax mexicanus*) in swim tunnels of the same size, and tested in flows at a similar range of speeds. The high flow speeds in both studies suggest that the flow gradients in these experiments were small. We assume that the lateral line feedback were equivalent in both studies, as the subjects were conspecific. Although the

tactile cues present in the experiments by *Bak-Coleman and Coombs, 2014* hinder our ability to reach a definitive conclusion on the effect of fish body size, the higher threshold speed observed in larger fish qualitatively supports our model prediction. The paucity of data for validation of the effect of fish size is a result of a lack of studies with matching experimental conditions, including dimensions of the flow facilities, flow conditions, and functionality of the lateral line.

In summary, we identified a total of five sets of experiments in support of our model, and nine sets of studies that offer inconclusive evidence. None of the data contradicted predictions from the proposed model.

Most experiments used in our comparison listed in *Appendix 3—table 1* were conducted in the past 25 years, and only two studies date back to before 1970. This disparity is attributed to an evolution of the methodologies for the study of rheotaxis over time. Among the earlier efforts, a large portion relied on observations of fish behavior in the field (*Arnold, 1974*). Although these studies minimized the introduction of external stimuli stemming from human presence and unfamiliar environments that could alter the behavior of fish in the wild, a lack of flexibility in the design of controlled experiments in the field, together with an insufficient measurement resolution, has led to only a limited number of works that could distinguish the impact of one sensory cue from another. As a result, a large number of earlier efforts do not meet our inclusion criteria.

Likely, the interest in fish rheotaxis was recently reignited owning to the advancements in technologies that could selectively deactivate specific fish sensory organs, thereby allowing for the targeted investigation of the role of each sensory cue in rheotaxis. For instance, pharmacological methods that could disable the lateral line led to studies (*Baker and Montgomery, 1999*; *Montgomery et al., 1997*) challenging the long-standing perception that the lateral line could not mediate rheotaxis. In addition, the development of high speed cameras with infra-red sensing capabilities enabled precise measurements of fish behavior in the dark, allowing for the elimination of visual cues from the study of lateral line functionality in rheotaxis.

Some early experiments that have been considered in the past as evidence against the role of the lateral line are not listed in *Appendix 3—table 1* due to a lack of a controlled experimental design in the field setting. For example, some species of salmonids, including salmon and trout, were observed to swim against the current in the day and rest on the bottom of a stream at night (*Davidson, 1949*; *Edmundson et al., 1968*; *Gibson, 1966*), leading to a conclusion that the lateral line played a minimal role in rheotaxis (*Arnold, 1974*). However, we did not include these experiments for our model validation due to confounding factors posed by the field settings, such as variations in water temperature (*Edmundson et al., 1968*; *Fraser et al., 1993*; *Needham and Jones, 1959*) and current speeds at different hours of the day. Daily fluctuations in the availability of food (*Waters, 1962*; *Elliott, 1965*) is another factor that could influence the activity levels of fish at night, as observed in white bass (*McNaught and Hasler, 1961*) and trout (*Elliott, 1965*). Another class of experiments that led to the previous rejection of lateral line was the demonstration of the imperative role of vision in rheotaxis. Experiments on salmon (*Hoar, 1954*) and herring (*Brawn, 1960*) showed a reduction in rheotaxis when vision was obscured in the dark or in muddy water, conflating the role of visual cues in rheotaxis. Again, these observations do not directly contradict the proposed model, which suggests that in the absence of visual cues, rheotaxis could still manifest provided that the flow speed is sufficiently high.

**Appendix 3—table 1.** Relevant publications on fish rheotaxis in the absence of visual cues, identified through literature review.

| Reference | Fish | | Swimming domain | Flow properties | | *Sensory cues | Rheotaxis threshold speed |
|---|---|---|---|---|---|---|---|
| | Species | Length | | Flow speed | Flow gradient | | |
| *Bak-Coleman et al., 2013* | Giant danio (*Devario aequipinnatus*) | 6.0–7.3 cm | Flow tank of $25 \times 25 \times 25\,\mathrm{cm}$ ($L \times h \times W$) | 0, 3, and $7\,\mathrm{cm/s}$ | Re $\sim 7500$ at LL+ threshold speed; flow gradient expected to be small near center of tank | LL+/LL- | $\leq 3\,\mathrm{cm/s}$ |

*Appendix 3—table 1 Continued on next page*

Appendix 3—table 1 Continued

| Reference | Fish | | Swimming domain | Flow properties | | *Sensory cues | Rheotaxis threshold speed |
|---|---|---|---|---|---|---|---|
| | Species | Length | | Flow speed | Flow gradient | | |
| *Bak-Coleman and Coombs, 2014* | blind cavefish (*Astyanax mexicanus*) | 4.2 –5.0 cm | Flow tank of 25 × 25 × 25 cm ($L \times h \times W$) | 0, 1, 2, 3, 4, 7 and 8 cm/s | $Re \sim 2000$ at LL+ threshold speed; flow gradient expected to be small near center of tank | LL+/LL-; fish made transient contacts with substrate | LL+: 0.90 cm/s; LL-:0.54 cm/s |
| [†]*Baker and Montgomery, 1999* | blind cavefish (*Astyanax fasciatus*) | 4 –7 cm | Flow tank of 51 × 9 × 20 cm ($L \times h \times W$) | 0, 2, 3, 5, 9 and 16 cm/s | $Re \sim 2000$ at LL+ threshold speed; flow gradient expected to be small near center of tank | LL+/LL-; tactile senses | LL+: 2–3 cm/s;LL-: 9–16 cm/s |
| *Elder and Coombs, 2015* | Mexican tetras (*Astyanax mexicanus*) | 8.3 cm | Flow tank of 25 × 25 × 25 cm ($L \times h \times W$) | 0, 1, 2, 4, 7, and 12 cm/s | $Re \sim 5000$ at threshold speed; flow gradient expected to be small near center of tank | LL+/LL- | $\sim 2$ cm/s for LL+ and LL- |
| *Kulpa et al., 2015* | blind cavefish (*Astyanax mexicanus*) | 4.4 –5.3 cm | Flow tank of 25 × 25 × 10 cm ($L \times h \times W$) | Maximum speed of 8 cm/s | Jet flow across center of tank; flow gradient expected to be large | LL+/LL- | $\leq 8$ cm/s |
| [‡]*Lyon, 1904* | blind Fundulus | unspecified | Trough with unspecified dimensions; tideway leading to pond | "not too strong current" in trough and current with "more or less eddy and irregularity" in tideway | Flow gradient expected to be small | LL+; some fish gained tactile senses | Not measured; rheotaxis elicited only by tactile cues |
| [‡]*Lyon, 1904* | blind Fundulus | unspecified | Trough with unspecified dimensions | flow "gushing rather violently" | Jet flow; flow gradient expected to be large | LL+ | Not measured; rheotaxis elicited by flow |
| [†]*Montgomery et al., 1997* | blind cavefish (*Astyanax fasciatus*) | 4 –7 cm | [§]Flow tank of 51 × 9 × 20 cm ($L \times h \times W$) | 0, 2, 3, 5, 9, and 16 cm/s | $Re \sim 2000$ at LL+ threshold speed; flow gradient expected to be small near center of tank | LL+/LL-; tactile senses | LL+: 2–3 cm/s; LL-: 9–16 cm/s |
| *Oteiza et al., 2017* | zebrafish (*Danio rerio*) larva 5–7 days post fertilization (dpf) | unspecified | 13 cm-long circular tube with diameter 1.27–4.76 cm | 0.2–0.8 cm/s | Low to high flow gradients identified through particle image velocimetry | LL+/LL- | LL+: rheotaxis observed as low as 0.2 cm/s |
| *Peimani et al., 2017* | zebrafish (*Danio rerio*) larva 5–7 dpf | estimated $\sim 0.35$ cm | Flow channel of 63.3 × 1.6 × 0.55 mm ($L \times h \times W$) | 0.95–3.8 cm/s | $Re \sim 10$ at threshold speed; flow gradient expected to be large | LL+ | 0.95 cm/s |
| *Suli et al., 2012* | zebrafish (*Danio rerio*) larva 5 dpf | $\sim 0.33$ cm | Flume of 110 × 3.7 × 2.8 cm ($L \times h \times W$) | 0.075, 0.15, 0.2 cm/s | $Re < 75$; flow gradientexpected to be large | LL+/LL- | Not quantified |

Appendix 3—table 1 Continued on next page

Appendix 3—table 1 Continued

| Reference | Fish | | Swimming domain | Flow properties | | *Sensory cues | Rheotaxis threshold speed |
|---|---|---|---|---|---|---|---|
| | Species | Length | | Flow speed | Flow gradient | | |
| *Van Trump and McHenry, 2013* | blind Mexican cavefish (*Astyanax fasciatus*) | 3 –7 cm | Cylindrical channel of $150 \times 11$ cm ($L \times D$) | 0, 1, 2, 4, 6, 8, 10, 13,**16 cm/s** | $Re > 2000$ at threshold speed; flow gradient expected to be small near center of tank | LL+/LL- | 2–4 cm/s |

*LL+: lateral line enabled; LL−: lateral line disabled
†Data are extracted from the same set of experiments
‡Two experiments are considered from the same paper
§Data are from *Baker and Montgomery, 1999*.

**Appendix 3—table 2.** Results of the bibliographical research on fish rheotaxis in the absence of visual cues, used to validate the proposed model.

| Reference | Fish species | *Evidence | Comparison with model | |
|---|---|---|---|---|
| | | | Supportive | Inconclusive |
| Within studies | | | | |
| *Effect of lateral line* | | | | |
| *Bak-Coleman et al., 2013* | Giant danio (*Devario aequipinnatus*) | No significant difference in fish heading angle against current was detected between LL+ and LL- | | × |
| *Bak-Coleman and Coombs, 2014* | blind cavefish (*Astyanax mexicanus*) | Rheotaxis threshold speed was slightly (but not significantly) lower in LL- condition | | × |
| *Baker and Montgomery, 1999* and *Montgomery et al., 1997* | blind cavefish (*Astyanax fasciatus*) | Rheotaxis threshold speed was significantly higher in LL- condition; fish received intermittent tactile senses | | × |
| *Elder and Coombs, 2015* | Mexican tetras (*Astyanax mexicanus*) | No significant influence of LL condition was detected on rheotactic performance | | × |
| *Kulpa et al., 2015* | blind cavefish (*Astyanax mexicanus*) | Significantly higher rheotaxis index in LL+ fish than LL- fish in jet stream | × | |
| *Oteiza et al., 2017* | zebrafish (*Danio rerio*) larva 5–7 dpf | Posterior lateral line ablation or chemical neuromast ablation severely reduced rheotaxis | × | |
| *Suli et al., 2012* | zebrafish (*Danio rerio*) larva 5 dpf | LL hair cell damage led to a significant decrease in rheotaxis; regeneration of LL hair cells restored rheotaxis | × | |
| *Van Trump and McHenry, 2013* | blind Mexican cavefish (*Astyanax fasciatus*) | In LL+ and LL-, fish exhibited statistically indistinguishable rheotaxis behavior | | × |
| *Effect of flow gradient* | | | | |

*Appendix 3—table 2 Continued on next page*

Appendix 3—table 2 Continued

| Reference | Fish species | *Evidence | Comparison with model | |
|---|---|---|---|---|
| | | | Supportive | Inconclusive |
| *Lyon, 1904* | blind Fundulus | In a flow with small gradient, rheotaxis was elicited only when fish received tactile cues; in jet flow with large gradient, rheotaxis was elicited by flow without tactile cues. Lack of data on statistical significance | | × |
| *Oteiza et al., 2017* | zebrafish (*Danio rerio*) larva 5–7 dpf | Rheotaxis of fish improved with increasing gradient magnitudes | × | |
| Across studies | | | | |
| *Effect of channel width* | | | | |
| *Bak-Coleman and Coombs, 2014*; *Baker and Montgomery, 1999* | blind cavefish (*Astyanax mexicanus*); blind cavefish (*Astyanax fasciatus*) | Significantly different threshold speed for LL+ fish: $0.90 \pm 0.137$ cm/s (mean ±s.e.m.) in 25 cm wide tunnel; between 2 cm/s and 3 cm/s in 9 cm wide tunnel. Tactile cues available to fish in *Bak-Coleman and Coombs, 2014* | | × |
| *Bak-Coleman and Coombs, 2014*; *Van Trump and McHenry, 2013* | blind cavefish (*Astyanax mexicanus*); blind cavefish (*Astyanax fasciatus*) | Significantly different threshold speed for LL+ fish: $0.90 \pm 0.137$ cm/s (mean ±s.e.m.) in 25 cm wide tunnel; between 2 cm/s and 4 cm/s in $\sim 11$ cm diameter tunnel. Tactile cues available to fish in *Bak-Coleman and Coombs, 2014* | | × |
| *Oteiza et al., 2017*; *Peimani et al., 2017* | zebrafish (*Danio rerio*) larva 5–7 dpf | Onset of rheotaxis in LL+ fish observed at flow speed 0.95 cm/s in 1.6 mm wide tunnel; rheotaxis observed in LL+ fish at flow speed 0.2 cm/s in 2.22 cm diameter tunnel | × | |
| *Effect of body length* | | | | |
| *Bak-Coleman and Coombs, 2014*; *Elder and Coombs, 2015* | blind cavefish (*Astyanax mexicanus*); Mexican tetras (*Astyanax mexicanus*) | Significantly different threshold speed for LL+ fish: $0.90 \pm 0.137$ cm/s (mean ±s.e.m.) for 4.2–5.0 cm long fish; $1.96 \pm 0.350$ cm/s (mean ±s.e.m.) for 8.3 cm long fish. Tactile cues available to fish in *Bak-Coleman and Coombs, 2014* | | × |
| Total | | | 5 | 9 |

*LL+: lateral line enabled; LL−: lateral line disabled

## Appendix 4

### Stability analysis of the dynamical system based on the A-dipole

#### Derivation of the turn rate equation

Different from *Equation 8*, the A-dipole model treats fish as a slender body, which rotates in response to a gradient in the cross flow along its body. As such, under the A-dipole paradigm, the hydrodynamic turn rate becomes

$$
\begin{aligned}
\Omega^{(A)}(\vec{r}_f, \theta_f) &= \hat{v}_f^{\perp} \cdot \left[ \nabla \left( \vec{u}_w(\vec{r}, \vec{r}_f, \theta_f) + \vec{u}_b(\vec{r}) \right) \big|_{\vec{r}=\vec{r}_f} \right] \hat{v}_f \\
&= \tfrac{\pi^3 \rho^2 v_0}{4h} \cot\left( \tfrac{\pi y_f}{h} \right) \csc^2\left( \tfrac{\pi y_f}{h} \right) \cos\theta_f + \tfrac{8 U_0 \epsilon}{h} \left( \tfrac{y_f}{h} - \tfrac{1}{2} \right) \sin^2\theta_f.
\end{aligned}
\tag{30}
$$

While the equation for the cross-stream motion of the A-dipole remains the same as *Equation 10a*, the governing equation for its orientation requires modification. Substituting *Equation 30* into *Equation 6b* yields

$$
\dot{\theta}_f = \tfrac{\pi^3 \rho^2}{4} \cot\left( \pi \left( \xi + \tfrac{1}{2} \right) \right) \csc^2\left( \pi \left( \xi + \tfrac{1}{2} \right) \right) \cos\theta_f + 8\alpha\xi \left( \sin^2\theta_f + \kappa \right),
\tag{31}
$$

which is analogous to *Equation 10b* for a T-dipole. *Equation 10a* and *Equation 31* constitute the governing equations of the nonlinear planar dynamical system for an A-dipole fish model.

#### Determination of the equilibria of the dynamical system

Similar to our analysis of the T-dipole, the equilibria of the dynamical system for the A-dipole can be determined by setting $\theta_f = 0$ and $\theta_f = \pi$ in *Equation 10a* and *Equation 31*. Analogous to *Equation 14*, the cross-stream location $\xi$ of the equilibria can be determined through a transcendental equation,

$$
\tfrac{\pi^3}{32} \cot\left( \pi \left( \xi + \tfrac{1}{2} \right) \right) \csc^2\left( \pi \left( \xi + \tfrac{1}{2} \right) \right) = \mp\beta^{(A)}\xi,
\tag{32}
$$

where the negative sign corresponds to $\theta_f = 0$ and the positive sign to $\theta_f = \pi$. Similar to the analysis of the T-dipole, here, we introduce a nondimensional parameter $\beta^{(A)} = \alpha\kappa/\rho^2$.

For downstream swimming ($\theta_f = 0$), there exists one or three roots to *Equation 32* depending on the value of $\beta^{(A)}$. For $\beta^{(A)}$ smaller than the critical value of $\beta^* = \pi^4/32$, $\xi = 0$ is the only solution, while for $\beta^{(A)} > \beta^*$, two additional roots symmetrically located with respect to the centerline emerge. On the other hand, for upstream swimming ($\theta_f = \pi$), there is only one root at $\xi = 0$.

#### Local stability analysis of the dynamical system

Local stability of the equilibria is studied by linearizing *Equation 10a* and *Equation 31* about the equilibria. Specifically, for $\theta_f = 0$ and $\xi = 0$, the state matrix reads

$$
A = \begin{bmatrix} 0 & 1 - \tfrac{\pi^2 \rho^2}{6} \\ 8\kappa\alpha - \tfrac{\pi^4 \rho^2}{4} & 0 \end{bmatrix}.
\tag{33}
$$

Stability requires that the determinant of $A$ is positive ($\det A > 0$), that is,

$$
\tfrac{1}{24} \left( 6 - \pi^2 \rho^2 \right) \left( \pi^4 \rho^2 - 32\alpha\kappa \right) > 0.
\tag{34}
$$

Since the first factor is always positive ($\rho \ll 1$), the inequality is fulfilled when $\beta^{(A)} < \beta^*$. Hence, the system is stable if $\beta^{(A)} < \beta^*$ and unstable if $\beta^{(A)} > \beta^*$.

For $\theta_f = 0$ and $\beta^{(A)} > \beta^*$, we identify two additional equilibria symmetrically located with respect to the centerline. The state matrix for these equilibria reads

$$
A = \begin{bmatrix} 0 & 1 + \tfrac{\pi^2 \rho^2}{12} - \tfrac{1}{4}\pi^2 \rho^2 \sec^2(\pi\xi) \\ 8\kappa\alpha - \tfrac{1}{4}\pi^4 \rho^2 (2 - \cos(2\pi\xi)) \sec^4(\pi\xi) & 0 \end{bmatrix}.
\tag{35}
$$

Stability requires that $\det A > 0$, that is,

$$\tfrac{1}{48}\left(12 - 3\pi^2\rho^2\sec^2(\pi\xi) + \pi^2\rho^2\right)\left(-32\alpha\kappa + \pi^4\rho^2(2 - \cos(2\pi\xi))\sec^4(\pi\xi)\right) > 0. \tag{36}$$

As demonstrated in Methods and Materials Section, the first factor in *Equation 36* can be assumed to be positive. Thus, the inequality in *Equation 36* is equivalent to

$$\beta^{(A)} < \tfrac{\pi^4}{32}(2 - \cos(2\pi\xi))\sec^4(\pi\xi), \tag{37}$$

which is always satisfied for any choice of $\beta^{(A)}$. Therefore, the additional equilibria emerging for $\beta^{(A)} > \beta^*$ at $\xi \neq 0$ are always stable.

For upstream swimming along the centerline ($\theta_f = \pi$ and $\xi = 0$), the state matrix is

$$A = \begin{bmatrix} 0 & \frac{\pi^2\rho^2}{6} - 1 \\ 8\kappa\alpha + \frac{\pi^4\rho^2}{4} & 0 \end{bmatrix}. \tag{38}$$

Similarly, stability requires that $\det A > 0$, that is,

$$\tfrac{1}{24}\left(6 - \pi^2\rho^2\right)\left(32\alpha\kappa + \pi^4\rho^2\right) > 0. \tag{39}$$

Since both the first and the second factors are positive, the condition in *Equation 39* is always satisfied and the equilibrium is stable.

Comparing predictions with the T-dipole, results are similar in that there is a critical threshold that modulates the stability of upstream/downstream swimming, with a preference for rheotactic behavior above the threshold. For the T-dipole, below the critical threshold, neither swimming downstream nor upstream along the centerline of the channel are stable, and above the threshold, the latter becomes stable. Put simply, at low flow speed it is difficult for fish to orient in the flow, but they can successfully orient against the flow at high speed. For the A-dipole, below the critical threshold, both swimming downstream and upstream along the centerline of the channel are stable, and above the threshold, only the latter remains stable. In other words, fish can orient in the flow at low speed, without a preference for downstream or upstream swimming, but at high speed only swimming upstream becomes feasible.

