## [Editor Report]

The authors present a simple model of fish swimming in a channel and reacting to the surrounding flow with their lateral line and no other sensory system. They demonstrate that the fish stably orients upstream in certain conditions. Particularly, rheotaxis can emerge even in the absence of sensory feedback, purely as a consequence of passive hydrodynamic interactions in the presence of the walls.

---

## [Decision Letter]

**Decision letter after peer review:**

Thank you for submitting your article "Hydrodynamic model of fish orientation in a channel flow" for consideration by *eLife*. Your article has been reviewed by 3 peer reviewers, and the evaluation has been overseen by a Reviewing Editor and Aleksandra Walczak as the Senior Editor. The reviewers have opted to remain anonymous.

Essential revisions:

1) While the attempt to compare predictions with data is clearly valuable, the data appear inconclusive and weaken the results. Please present these data as a supplementary information, de-emphasize the comparison and temper claims about biological relevance.

2) Please discuss the assumptions of the model, and in what parameter regime they are expected to hold. in particular, please provide evidence for adequacy of the dipole model; the rationale for modelling the external flow as a constant + parabolic perturbation; and the rationale for the feedback.

3) Please validate the model, for example with numerical simulations that solve the Navier Stokes equations.

4) Please include a discussion about experiments that could definitively test the hypothesis with real fish, even if you will not perform them.

*Reviewer #1 (Recommendations for the authors):*

The paper is missing an independent measure of function. Another line of evidence that this phenomenon is actually contributing to rheotaxis would lend strength to their argument. The strength of the theoretical work largely outweighs the support from the biological literature.

As of now, they come up with a model that matches some of what has been shown in the (inconclusive) literature. Open loop robotics is a useful suggestion and should be implemented for conclusive results. At the very least, I would have the authors suggest more concrete experiments, even if they will not do them. This shows a commitment to advancing our understanding of the actual biological phenomenon. Regarding open loop robots, and I would like to see a more detailed description of what results they would expect and how this would strengthen their argument. Likewise, a discussion on how wider flumes are expected to increase rheotaxis should be included, rather than simply mentioning that this is supported in some biological studies.

Table 1 summarizes experiments for rheotactic fish without vision, with some studies supportive and most inconclusive. I found this table marginally useful (the fault of the studies, not the authors) because these experiments have other sensory modalities intact that can generate rheotaxis. To use this to support that fish can use only their lateral line for rheotaxis is misleading, in my opinion. For the importance of a single parameter threshold to be realized, one would observe cross stream sweeping movements at higher flows for a good majority of fish swimming studies, and this is simply not the case. Literature could be better interpreted. Rheotaxis is a multi-modal behavior, so ablation of the lateral line is not sufficient to guarantee that rheotaxis can occur without it.

If improving biological experiments is the goal, then more discussion is warranted. A discussion of reafference and what flow information is incorporated in the model-acceleration, velocity, both? What do larval zfish (superficial neuromasts) behaviors tell us that are consistent (or not) with adult behaviors (canal neuromasts)?

The impact contracts in the face of what can be conclusively supported, leaving the authors to describe their work as rectifying a methodological oversight in laboratory experiments.

*Reviewer #2 (Recommendations for the authors):*

Please conduct a quantitative validation and sensitivity analysis of the model.

*Reviewer #3 (Recommendations for the authors):*

I have now read this manuscript on the stability of a fish in a channel flow. As stated in my public review, I think the assumptions of the model are not adequate for the problem studied. For this reason, I believe the manuscript is not publishable as it is. Here is a list of some comments.

- In the introduction, one aspect of fish locomotion seems to be missing: proprioception. Fish sense their surrounding also be sensing how the hydrodynamic forces and moments affect their own shape.

- The dipole model is very crude, is there any experimental evidence that this model would be appropriate to model zebrafish larvae, for instance? I believe not, because zebrafish do not swim continuously and the Reynolds number is far too small for a potential flow assumption. Besides, there is a wake behind a swimming fish that is not captured by the dipole model.

- What is the Reynolds number of the channel flows in the experiments? I believe that superposing a constant flow and a parabolic flow is not the best model for these channel flows. Usually, people use some sort of plug flow instead.

- It is strange to have a mix between a viscous flow for the channel flow and potential flow for the swimmer. What is the rationale for this mix?

- Eq. (8) describes how the potential flow rotates the dipole. I understand that you consider a sort of dumbbell model, with the dumbbell oriented perpendicular to the swimming direction. But fish are elongated in the longitudinal direction, so it seems strange.

- Eq. (9) is not a "rich nonlinear dynamics" as the author state, but simply a linear feedback with the distance from the wall. Most visual models have the same kind of feedback.

- The comparison with experimental results is presented in tables. This is not very clear and the reader is left with the impression that most comparisons are inconclusive.

---

## [Author Response]

Essential revisions:1) While the attempt to compare predictions with data is clearly valuable, the data appear inconclusive and weaken the results. Please present these data as a supplementary information, de-emphasize the comparison and temper claims about biological relevance.

Thank you for the constructive feedback. We acknowledge that most data in the literature only provide inconclusive evidence and have followed your advice to move the data to Appendix 3 to avoid over-emphasizing the biological relevance. Furthermore, we have largely rewritten the Discussion section, focusing on the model results and acknowledging limitations inherent to any comparison with available empirical findings.

2) Please discuss the assumptions of the model, and in what parameter regime they are expected to hold. in particular, please provide evidence for adequacy of the dipole model; the rationale for modelling the external flow as a constant + parabolic perturbation; and the rationale for the feedback.

Thanks for the great suggestion, which has prompted us to significantly expand on the theoretical contribution of the work by performing an original computational validation of the dipole model – which has never been done in the literature. Numerical results are included in the new subsection Numerical Validation of the Dipole Model in the Results section and accompanying Appendix 2. With respect to the rationale for selecting a constant flow + parabolic perturbation, we have added details in the main text. Our assumptions on the flow profile consisting of a uniform flow (plug flow) with a parabolic profile is the simplest form that approximates the mean flow while incorporating vorticity near the channel centerline. Importantly, should one account for more complex velocity profiles, the linear stability analysis will not change, as it is based only on local flow features at the centerline. As a result, our claims regarding the stability of rheotaxis above a critical threshold speed would not be affected by the assumption of a more complex velocity profile. In the same vein, a nonlinear feedback mechanism would not change the results from the linear stability analysis. A paragraph has been added after Equation (22) in the Methods and Materials section.

3) Please validate the model, for example with numerical simulations that solve the Navier Stokes equations.

Once again, thanks a lot for the request that prompted us to undertake the computational simulation of Navier Stokes equations of a fish undulating its body in a channel flow. Numerical simulations demonstrate that the dipole model provides a sufficiently accurate approximation of the mean flow field, capturing the flow circulation from the fish head to the tail and the elongated streamline pattern near the walls. (See the new Numerical Validation of the Dipole Model subsection in the manuscript, with additional details provided in Appendix 2.)

4) Please include a discussion about experiments that could definitively test the hypothesis with real fish, even if you will not perform them.

Thank you for the constructive request. We believe that the best experimental approach to test the hypothesis of passive rheotaxis should actually involve robotic, rather than real, fish. Specifically, robotic fish mimicking the locomotory patterns of live animals and working in open loop (without any sensory input) could be used to test the hypothesis that above a critical flow speed, rheotaxis is stable. We have added two paragraphs in the Discussion section (paragraphs five and six) to detail an experiment that could definitively test the hypothesis of a completely passive hydrodynamic mechanism for rheotaxis. Therein, we also expand on the limitations of existing methods to disable sensory modalities in live animals. We specifically borrow arguments from Coombs et al. (Journal of Experimental Biology, 2020, 223, 223008) to articulate that an attempt to block all sensory modalities may challenge fish ability to swim; for example, disabling vestibular senses may cause inability to maintain an upright posture (see paragraph four of the Discussion section).

Reviewer #1 (Recommendations for the authors):The paper is missing an independent measure of function. Another line of evidence that this phenomenon is actually contributing to rheotaxis would lend strength to their argument. The strength of the theoretical work largely outweighs the support from the biological literature.As of now, they come up with a model that matches some of what has been shown in the (inconclusive) literature. Open loop robotics is a useful suggestion and should be implemented for conclusive results. At the very least, I would have the authors suggest more concrete experiments, even if they will not do them. This shows a commitment to advancing our understanding of the actual biological phenomenon. Regarding open loop robots, and I would like to see a more detailed description of what results they would expect and how this would strengthen their argument. Likewise, a discussion on how wider flumes are expected to increase rheotaxis should be included, rather than simply mentioning that this is supported in some biological studies.

We agree with the Reviewer on the limited support in the biological literature. In the revised manuscript, we discuss limitations of the existing literature to validate our model (see the fourth paragraph of the Discussion section). We have elaborated on an experimental protocol involving robotic fish operating in open loop that could be performed to validate our model (see paragraphs five and six of the Discussion section). In addition, we have included a more detailed discussion of the role of the channel width (see paragraph eight of the Discussion section).

Table 1 summarizes experiments for rheotactic fish without vision, with some studies supportive and most inconclusive. I found this table marginally useful (the fault of the studies, not the authors) because these experiments have other sensory modalities intact that can generate rheotaxis. To use this to support that fish can use only their lateral line for rheotaxis is misleading, in my opinion. For the importance of a single parameter threshold to be realized, one would observe cross stream sweeping movements at higher flows for a good majority of fish swimming studies, and this is simply not the case. Literature could be better interpreted. Rheotaxis is a multi-modal behavior, so ablation of the lateral line is not sufficient to guarantee that rheotaxis can occur without it.

We thank the Reviewer for the insightful comments on the interpretation of the literature in our manuscript. We agree that most of the studies identified in the literature do not establish an idealized condition where only the lateral line was intact while all other sensory modalities were eliminated, leading to “inconclusive” comparisons for most studies. In the revised manuscript, we have softened and focused all of the claims, moved the validation against the literature to Appendix 3, and have added context to acknowledge limitations of the existing experimental studies (see, for example, paragraphs four and eleven of the Discussion section).

If improving biological experiments is the goal, then more discussion is warranted. A discussion of reafference and what flow information is incorporated in the model-acceleration, velocity, both? What do larval zfish (superficial neuromasts) behaviors tell us that are consistent (or not) with adult behaviors (canal neuromasts)?

The model does not incorporate flow information from acceleration. Information from velocity is encapsulated in the form of the background flow circulation that is sensed through the lateral line, and, original to this work, the bidirectional interaction of the fish with the confined flow. We clarified these aspects when introducing the model, softened biological statements throughout the text, and added additional discussion regarding the model limitations (see the second-to-last paragraph of the Discussion section).

The impact contracts in the face of what can be conclusively supported, leaving the authors to describe their work as rectifying a methodological oversight in laboratory experiments.

In the text, we acknowledged the limited support of the literature and highlighted the value of the conclusions that can be drawn on the basis of a pure fluid mechanics perspective.

Reviewer #2 (Recommendations for the authors):Please conduct a quantitative validation and sensitivity analysis of the model.

Our two-dimensional numerical validation model was developed by extracting the geometry and body motion of a giant danio from the literature. The numerical model was implemented into a laminar flow channel whose speed was selected such that the net drag on the model was zero. The local and global flow features generated by the numerical fish model were compared with those produced by a dipole model with identical swimming speed. (See the new Numerical Validation of the Dipole Model subsection, with additional details in Appendix 2.)

Reviewer #3 (Recommendations for the authors):I have now read this manuscript on the stability of a fish in a channel flow. As stated in my public review, I think the assumptions of the model are not adequate for the problem studied. For this reason, I believe the manuscript is not publishable as it is. Here is a list of some comments.- In the introduction, one aspect of fish locomotion seems to be missing: proprioception. Fish sense their surrounding also be sensing how the hydrodynamic forces and moments affect their own shape.

We thank the Reviewer for highlighting this. We have amended the Introduction to include a more comprehensive list of the sensing modalities employed by swimming fish. (See the added text in the second paragraph of the Introduction.)

- The dipole model is very crude, is there any experimental evidence that this model would be appropriate to model zebrafish larvae, for instance? I believe not, because zebrafish do not swim continuously and the Reynolds number is far too small for a potential flow assumption. Besides, there is a wake behind a swimming fish that is not captured by the dipole model.

In the revised manuscript we have clarified that the entire analysis was not tailored to explore zebrafish larvae motion, which is indeed at a much lower Reynolds number. Furthermore, we have added to the manuscript a comparison of the dipole model with the results from a twodimensional computational fluid dynamics simulation of the flow around a swimming giant danio. Simulation results demonstrate that the dipole model is successful in replicating the global, mean flow features. (See the new Numerical Validation of the Dipole Model subsection, with additional details supplied in Appendix 2.)

- What is the Reynolds number of the channel flows in the experiments? I believe that superposing a constant flow and a parabolic flow is not the best model for these channel flows. Usually, people use some sort of plug flow instead.

The Reynolds numbers for the experiments, which are reported in Appendix 3 Table 1, span a broad range from 10 to 10,000. At the low end of the range, a parabolic velocity profile is expected. At the higher end of the spectrum, the velocity profile is expected to be turbulent, resembling a top hat (plug flow). In both cases, near the channel centerline there will be a degree of shear flow, offering some bias to the animal by which it can appraise the flow environment and distinguish downstream from upstream. This is the only information that we utilize in our model and its local stability analysis to demonstrate the existence of a critical flow speed above which fish will perform rheotaxis. (See added text in the Results section after Equation (4) and in the Methods and Materials section after Equation (22).)

- It is strange to have a mix between a viscous flow for the channel flow and potential flow for the swimmer. What is the rationale for this mix?

We apologize for the lack of clarity and have revised the manuscript to rectify this. We are not modeling the channel flow as purely viscous, as this would necessitate satisfaction of the noslip boundary condition (zero velocity on the walls). By superimposing a parabolic (rotational) perturbation to a uniform (irrotational) mean flow we offer an approximation of a turbulent velocity profile with the modest vorticity needed for fish hydrodynamic sensing. This approximation only employs one parameter to capture curvature of the velocity profile near the channel centerline. (See the added text in the Results section after Equation (4).)

- Eq. (8) describes how the potential flow rotates the dipole. I understand that you consider a sort of dumbbell model, with the dumbbell oriented perpendicular to the swimming direction. But fish are elongated in the longitudinal direction, so it seems strange.

We thank the Reviewer for the comment. There are two dipole models for swimming fish in the literature, the so-called A- and T-dipoles. Our model employs the T-dipole, which is consistent from the standpoint of vortex dynamics for the propagation and orientation of a vortex pair (see Equation (12) and associated discussion in the manuscript). In the A-dipole, in contrast, the fiducial points governing the turning of the dipole are along the swimming direction and thus not coincident with the driving vortices. Whether the A- or T-dipole is a better model for a fish is a subject for debate, which we aim to address through the proposed experiments using robotic fish. We note that one interpretation of the T-dipole is that it treats the fish as a bluff body, which rotates according to a gradient in the pressure drag across the width of its body. This torque is amplified by a mismatch in the hydrodynamic drag experienced by the paired pectoral fins of fish. (See the text added in the Derivation of the Turn Rate Equation for Fish Dynamics subsection after Equation (13).)

- Eq. (9) is not a "rich nonlinear dynamics" as the author state, but simply a linear feedback with the distance from the wall. Most visual models have the same kind of feedback.

We apologize for the overstatement and have revised the terminology in the manuscript.

- The comparison with experimental results is presented in tables. This is not very clear and the reader is left with the impression that most comparisons are inconclusive.

We acknowledge that most data in the literature only provide inconclusive evidence. In the revised manuscript, we have moved the literature review to Appendix 3 to de-emphasize the biological statements. Therein, we also reorganized the text into paragraphs to facilitate the interpretation of existing literature in the context of the proposed model.

[Editors' note: we include below the reviews that the authors received from another journal, along with the authors’ responses.]

Reviewer #1 (Remarks to the Author):This is an interesting and potentially significant modelling evaluation of the possibility of rheotaxis behaviour in the absence of sensory information. However, several aspects are problematic need to be clearly addressed and/or clarified:1. In the discussion, the authors quote Lyon (1904): “It is equally absurd to imagine a fish in the Gulf Stream to be stimulated and oriented by a uniform forward motion of the water. Whether orientation be a simple reflex or a conscious process, points of reference – i.e., points relatively at rest – are necessary.” And they go on to say “It is the gradient of the flow that creates hydrodynamic points of reference for a fish to undertake rheotaxis in the dark, even without access to sensory information through the lateral line”. The importance of this flow non-uniformity needs to be more apparent in the summary. It needs to be clearly stated that in uniform flow “points relatively at rest are necessary” but that in non-uniform flow other options may be available.

Thank you for the suggestion. We modified the text accordingly.

2. The summary makes the claim: of ”the existence of a critical flow speed for fish to successfully orient without access to sensory information”. And the introduction expands on this…….”While offering an elegant pathway to explain rheotaxis, the framework of Oteiza et al. does not include a way for rheotaxis to be performed in the absence of information about the local vorticity field. Several experimental studies have shown that fish can perform rheotaxis even when their lateral line is partially or completed ablated, provided that the flow speed is sufficiently large. There are at least 2 problems here.Firstly, the authors have not adequately examined the biological literature (which I admit is complex). Most (if not all) of the studies in the review they cite, that show rheotaxis in LL- fish apply to fish that are benthic, or maintain contact with the base of walls of the tank, which of itself provides tactile means of maintaining rheotaxis. The rheotaxis thresholds in the Coombs etc. table most likely occur at a threshold for slip detection.

We expanded on the biology literature and commented on the possibility that fish with a compromised lateral line may rely on tactile sensing. However, the claim of the Reviewer that the “rheotaxis thresholds in the Coombs etc. table most likely occur at a threshold for slip detection” is a conjecture, which we find unlikely, since rheotaxis in the absence of visual cues therein is accompanied by periodic cross-section sweeping -- a locomotory pattern also observed in a midwater swimming species, giant danio (Bak-Coleman et al., 2013, JEB 216, 4011-4024) -- which would be difficult to justify through slip detection. Should rheotaxis be a sole outcome of slip detection, the fish would either stay in place or erratically swim in the channel, without exhibiting the regular cross-sweeping motion that is observed.

Through our literature review, we identified a total of eight experiments that explored the role of lateral line in rheotaxis, as shown in Table 1 in the Results section. These experiments were conducted either using blind fish or fish in the dark, such that visual cues were not accessible. We intentionally excluded pleuronectiform flatfishes, including the benthic fish plaice, which were frequently referred to by Coombs et al. (see the Methods Section for the exclusion criteria). Among the identified studies, we found three in support of our model, showing a significant decrease in rheotaxis performance when the lateral line was disabled. The rest were found to be inconclusive, where fish were observed to either receive tactile information or failed to exhibit statistically significant differences in threshold speed. No dataset was found to contradict our model. Through our literature review, we also discovered that the rheotaxis threshold speeds for zebrafish larvae and blind cavefish were larger in thinner channels. If the rheotaxis threshold was solely determined by tactile senses, we should expect the threshold speed to be independent of the channel width.

Secondly, and perhaps more problematic for this manuscript is that the example they use in Figure 1 as their illustration of the problem is in fact a LL+ fish (see Elder, J. and Coombs, S. The influence of turbulence on the sensory basis of rheotaxis. Journal of Comparative Physiology A 201, 667–680 (2015) the start of Figure legend 7 “Figure 7 Behavioral results from a single LL+ fish exposed to the TGS+ condition in the dark at 4 cm/s current speed).

We confirm that we used Figure 7 in which fish have a lateral line. The only purpose of using Figure 7 was to illustrate the concept of cross-sweep motions during rheotaxis. That being said, rheotactic accuracy and index are marginally affected by the presence of the lateral line as evinced by comparing the black curves in Figures 4b and 4d with those in 4a and 4c of the same paper, respectively. We apologize if we have given an incorrect impression and, in agreement with Reviewer 2, we opted for a schematic rather than showing experimental data.

Also, the boundary layer profile, shown in (a) in this figure doesn’t adequately represent the profile for these flow speeds in this tank size, and indeed the example they refer to has a turbulencegenerating structure present.

We reiterate that Figure 1a was intended as a schematic of a fish swimming in a generic channel and 1b as a representative illustration of the phenomenon based on the literature (this was indeed among the very few trajectory data publicly available).

3. How does the model reflect normal fish experiments (flows, gradients, habitats etc)? The use of a non-dimensionalized model may well be fully appropriate, but does make relevant comparisons between behavioural experiments and the model non-trivial. Without a more detailed description of how the model equates to observed behaviour, it is difficult to agree with the discussion statement “In agreement with experimental observations, we uncovered an equilibrium at the channel centerline for upstream swimming whose stability is controlled by a single nondimensional parameter that summarizes flow speed, flow gradient, lateral line feedback, fish size, and channel width. Above a critical value of this parameter, rheotaxis is stable and fish will begin periodic cross-stream sweeping movements whose amplitude can be as large as the channel width.” This is particularly so as the comparisons are again referenced to rheotaxis thresholds that are thought to be a result of edge contract and slip detection. Without further justification and biologically relevant validation of the model I am unable to agree with the conclusions stated in this paper. So in summary, I think the model is potentially interesting, but given the concerns raised above do not think that it takes us beyond the Oteiza et al. “elegant hydrodynamic mechanism for fish to actively perform rheotaxis”, and the currently understood complexity of multisensory contributions, including vision, lateral line, tactile, and vestibular sensing.

Thank you for the detailed feedback. In addition to reporting our main claim in nondimensional form, we included results in dimensional form along with a detailed comparison with specific biology literature. In the Results section, we explicitly expressed the fish rheotaxis threshold speed as a function of its lateral line feedback, flow gradient, swimming domain size, and fish body length. At the time of the initial submission, we focused the presentation on the model development more than its biological value. We improved the biological relevance of our study and performed an extensive bibliographical survey that resulted in a database of experimental results that help validate our claims and better frame them in the biology literature on multisensory orientation. For example, the collected data indicate that the rheotactic threshold increases with the size of the fish and decreases with the curvature of the flow and the ablation of the lateral line: all these results are in agreement with our theory. To address the concern raised by the Reviewer on the reference to benthic fishes that depend on tactile senses to perform rheotaxis, in these tables we excluded studies where the subjects were benthic flatfish and considered “inconclusive” studies in which tactile cues were available.

Reviewer #2 (Remarks to the Author):Porfiri and Peterson describe a mathematical model of rheotaxis in swimming animals. Their calculations suggest that fish may orient swimming against flow through an interaction between the self-generated dipole flow field from swimming and spatial gradients in the free stream flow.I found this paper difficult to follow and I therefore do not understand the reasoning that underlies their authors’ arguments that they have resolved a centuries-long mystery about rheotaxis. I have itemized some of the instances were the authors have failed to define key terms, did not offer a justification for their decisions for the model design, and did not offer a clear basis for their statements of findings. Someone with a fluid dynamics background who is familiar with related models could probably decode the meaning of this study from the equations alone, but the broad readership of this journal requires a more clear explanation.

We apologize for not having made the model more readily digestible to a broad readership. We revised the language to improve readability for people outside fluid dynamics and dynamical systems.

Specific CommentsL30 – Hair cells do not “comprise” the lateral line system. They are essential components, but the lateral line includes other features (e.g. neuromast cupulae and afferent and efferent neurons).

Thank you for the suggestion. We clarified the text accordingly.

L54 – “different answers” — Different from what?

We deleted the confusing statement.

L55-63 – It is not clear to me what specific intent there is that motivates this modeling. A statement of a hypothesis, question, or aim would help.

The proposed model aims to unveil the implications of a fish being an invasive sensor that actively augments the background flow and responds to it. We clarified throughout the manuscript its objective.

L81 – Please explain on what basis the wall may be modeled by the mirror image of the fish. It might help to include streamlines (perhaps in gray) for the mirror-image fish in Figure 2.

The method of images is a classical method of treating walls in potential flows; we added a reference to the text in the Results section.

Please explain where the boundary layer flow depicted in Figure 1a is designed by the equations.

We clarified this in the caption of Figure 1.

L112 – Please explain the purpose of equations 6a and 6b.

We amended the text to improve its clarity.

L115 – I am not sure that I follow the logic here. I would expect greater resistance to swimming in the center of the channel.

We clarified the physical explanation behind equation 7. We note that the expression does not represent "drag" on the fish in the traditional sense of viscous flows, which would depend on the relative speed of the fish with respect to the background; rather, it is a non-viscous effect from the interaction with the walls.

L118+ – Please explain the meaning of “Hydrodynamic turn rate is incorporated...” What is HTR? What is it incorporated into?

We have added explanation for this at the beginning of the subsection, as detailed in our reply above.

L122 – On what basis? Through an algorithm of motor control, or by passive dynamics?

This is entirely passive hydrodynamics. In this sentence, we have changed “rotate” to “be rotated” to emphasize it is passive dynamics.

L128 – Please define “hydrodynamic feedback”

We added a definition to the sentence.

L132 – Please give the justification for these calculations.

We added justifications for the need to linearize dynamics in the study of stability.

L137 – Please use words to explain the meaning of “the right hand side of (6)”.

We added text.

L192 – I am not sure I agree that the present model resolves this dilemma: doesn’t the model assume boundary layer flow, instead of the laminar flow referred to in the quote? I also don’t follow the meaning of “hydrodynamic points of reference”.

The model does not assume boundary layer flow: we have a uniform (potential) flow with zero boundary layer thickness on which we superimpose a small spatial gradient to the mean flow that is reminiscent of a “laminar flow”. We reworded the text pertaining to hydrodynamic points of reference explaining that these are regions with distinctive flow features from the background.

Figure 1b – I do not think it is necessary to publish a figure from another study here, particularly because that data is not presently used.

We agree with the Reviewer and have replaced the figure with a schematic, which also addresses the concerns of Reviewer 1.

Reviewer #3 (Remarks to the Author):Key results: The authors devise a mathematical model that explains key features observed in a common behavior in fishes, the ability to orient into a current (rheotaxis). Their demonstration that active manipulation of the fluid, through coupled Fluid-Structure Interactions, can overcome sensory deprivation is novel and of broad interest.Validity: The major flaw as I see it is in a misrepresentation, albeit not intentional, of the biological literature. The basis for their systems dynamics explaining the “puzzling aspect” of rheotaxis, the need for a critical flow speed in order for rheotaxis to operate absent of sensory stimuli, is based ultimately on one paper, included in their cited review but tracked down here as Bak-Coleman et al. JEB 2013. In Bak-Coleman’s paper, fish had a block of the lateral line, but not of other sensory modalities that can contribute to rheotaxis. Most importantly, the vestibular system in the inner ear can sense changes in body acceleration, which would certainly be the case if the onset of faster flow would case the animal to surge backwards. This would elicit a compensatory movement forward, temporarily or otherwise (it is hard to tell because this level of experiment is difficult to achieve). At the very least, the authors would need to revise their statement on how their model can account for upstream movement, not in the absence of all sensory stimuli, but only in the absence of lateral line stimuli. This distinction is made in line 198, but missing in other sections, misleading the reader to think that fish can perform rheotaxis in the absence of ANY sensory cues (line 15, 27). This sets up a strawman hypothesis. The authors should acknowledge the contribution of the vestibular system, potential auditory cues that accompany faster flows, and the presence of pressure sensors along the skin and fins of fishes. This would take the form of dorsal root ganglion cells and Rohon Beard cells, as well as sensory afferents in the fins, as documented by Melina Hale’s group. Currently, there is no biological experiment where all stimuli from rheotaxis can be abolished.

We apologize for the unintentionally misleading statements that could send mixed messages to the reader regarding the mechanisms underlying rheotaxis. We carefully revised the text and amended the Results and Discussion sections to clarify the possible involvement of other sensory cues in rheotaxis, including vision, vestibular system, and pressure sensing afferents in the fins.

That being said, the authors have demonstrated a unique contribution of bi-directional coupling to ease the need for sensory inputs. I am just doubtful that it has the relevance they propose for real biological systems. I would be more excited about their contributed if they limited it to a purely theoretical paper. Additionally, their claim that the onset of another behavior, crossstream sweeping movements, is not an inherent part of the rheotactic response as they proclaim. This could be an exploratory behavior observed in the one study they cite (and could be a speciesdependent behavior), but the long history of rheotaxis work does not include this as part of this innate behavior, which is simply the orientation of the fish upstream in the presence of a current. I found the writing vague and over-stepping at points.

Thank you for the excellent suggestion. We emphasized the paper contribution as a theoretical investigation of the effect of bidirectional coupling between fish and the surrounding flow, apt at demonstrating the potential consequences of constraining a fish in a water channel to study the principles of rheotaxis. At the same time and compatibly with the requests of the other Reviewers, we focused the writing to avoid overstepping and reinforced the biological framing of the paper. We acknowledge the Reviewer’s comment that cross-stream sweeping is not an inherent locomotory pattern of rheotaxis, and we have revised the text to indicate that this behavior has only been observed in some fish species (for example, cavefish and giant danio). We believe that now the paper makes a compelling case for an overlooked phenomenon that has never been studied (bidirectional interactions), irrespective of the senses that fish may use during orientation.

Originality and significance: The conclusion that Fluid Structure Interactions can obviate the need for sensory inputs is original and very interesting. However, the claim that the behavior of the real animal occurs in the absence of all sensory stimuli is not well-supported, and thus the house of cards crumbles and the interest level to the field wanes.

By refocusing the paper on the bidirectional coupling as the Reviewer suggests, we believe this concern has been addressed. Whether a fish uses none, one, or multiple senses, bidirectional interactions with the surrounding fluid flow remain: our paper demonstrates the consequences of this coupling. As detailed in our reply to your first comment, we amended the Results and Discussion sections to comment on the potential influence of other sensory modalities, including vision, vestibular system, and tactile sensors on the fish body surface.

Data and methodology: The model itself seems appropriate, that hydrodynamic feedback is related to the local circulation around a fish. Small assumptions that could lead to biologically disparate results include (1) a linear hydrodynamic feedback mechanism and (2) neglecting the inertia of the fish (fish cannot respond instantaneously to changes in fluid flow).

We added a paragraph in the Discussion section to acknowledge the main limitations of the model, including those you have identified.

Appropriate use of statistics and treatment of uncertainties: No statistical tests were conducted. Inclusion of a statistical treatment of the occurrence of the cross-stream behavior (from the original biological paper) modeled in Figure 1 would be especially useful.

Based on feedback from other Reviewers, we replaced Figure 1b with a schematic. In this revision, we conducted an extensive literature review and collated a table of experimental results that help validate our claims and frame them in the context of other sensory modalities. We identified several datasets indicating that the rheotactic performance decreases with the ablation of the lateral line and that the rheotaxis threshold decreases with the curvature of the flow and width of the channel -- all in agreement with our theory.

Conclusions: The model itself is of interest, but its relevance to the biological example selected is problematic. This is discussed elsewhere in this review, but an example of over-stating the problem is found in line 189 in Lyon’s quote. Reference points could be visual, magnetic or hydrodynamic (not just hydrodynamic). Again, in line 218 the authors describe a “counter-intuitive” conclusion of active manipulation, but this has long been discovered for the lateral line wallfollowing abilities of the blind cavefish.

Not only did we reinforce the biological basis, but also we refocused the work on the fundamental problem of bidirectional coupling as you suggested. We amended the text following the quotation of Lyon to resolve the over-statement by pointing out possible points of references.

Suggested improvements: I would suggest that the premise of the manuscript be altered to de-emphasize the biological relevance of the work. The argument that fish can orient upstream without the contribution of sensory information is built on shaky ground. My advice would be to re-orient the paper to ask how the model eases the task of sensory computation, rather than proposing a purely physical method of rheotaxis inherent in Fluid-Structure Interactions. Furthermore, a clearer distinction should be made on how this work builds on slender body theory of Lighthill and Taylor. The distinction from Oteiza’s work, which requires flow gradient from walls to make left-right comparisons, could be clearer (particularly Equation 7, line 115—the “retarding effect of the wall”).

We are deeply appreciative of the suggestion, which we have followed (compatibly with the requests by the other Reviewers), along with providing more details on the model. We emphasize that the current minimal model is built upon the finite-dipole framework, which does not rely on the slender body assumption.

References: Line 38—Several experiments are cited as evidence that rheotaxis occurs in the absence of lateral line information—this is a review paper that specifically demonstrates that rheotaxis is a multi-modal behavior, so ablation of one sensory modality is not sufficient to guarantee that rheotaxis can occur in the absence of sensory inputs. Several biological references should be included along the ones from Coombs—among them the authors Montgomery, Windsor, McHenry, Van Netten ,Chagnaud, Bleckmann, Hale, Liao, Trapani, Hudspeth, Kindt and Raible—each of these authors develops work on the lateral line or other relevant sensory modality that would influence these authors and their understanding of how the lateral line and other senses can be effectively ablated.

We strengthened the biological relevance of the work through an extensive literature review, as detailed in the Results and Methods Section. We identified several studies that demonstrated the role of lateral line in the absence of visual and tactile senses, as summarized in Table 1.

Treating the fish as an “invasive” sensor is an important approach, and I agree to the importance of this addition (line 48), but I found line 58 and 61 vague in terms of what this current manuscript offers. Some specificity would be useful here.

The proposed model unveils the crucial role of bidirectional coupling between fish and the surrounding fluid in rheotaxis, which has been overlooked in existing experimental and theoretical frameworks for the study of fish rheotaxis. We clarified the contribution of the manuscript in the Abstract and Discussion sections of the revised manuscript.

Clarity and context: The abstract, Intro and Discussion is largely accessible, but over-reaches. The manuscript is one where a good deal of theoretical work has been done that supports some observations during biological rheotaxis, of which there could be more parsimonious reasons other than the one the authors champion.

Thank you for the comment. We believe that addressing the comments above has provided a clearer focus for the proposed model, along with a stronger biological context upon which claims can be drawn.

Response to the second round of review

Reviewer #1 (Remarks to the Author):As stated in my original review “This is an interesting and potentially significant modelling evaluation of the possibility of rheotaxis behaviour in the absence of sensory information. However, several aspects are problematic need to be clearly addressed and/or clarified”. In my view the author’s rebuttal does not address these issues and in some respects exacerbates them.So I still disagree with the some of the fundamental claims of this manuscript:The abstract states: “we make an essential step to elucidate the hydrodynamic underpinnings of rheotaxis” Whereas, in my view, the manuscript does not elucidated the hydrodynamics of rheotaxis per se, – but shows the theoretical possibility of bidirectional coupling as a possible orientation mechanism under conditions of no sensory input. It is clear from the modelling, that “a fish as an invasive sensor” may theoretically face upstream and move from in a sweeping motion. But the biological arguments to support the statement that fish, in fact, do this are flawed.The abstract further states that their proposal “captures many of the puzzling aspects of rheotaxis, from the existence of a critical flow speed for fish to successfully orient, to the onset of crossstream sweeping movements while swimming against the flow observed in some fish species”The statement below quoted from Bak-Colman et al. 2013 contradicts this. The sweeping movements cease with lateral line deprivation. “When deprived of vision, fish move further upstream, but the angular accuracy of the upstream heading is reduced. In addition, visually deprived fish exhibit left/right sweeping movements near the upstream barrier at low flow speeds. Sweeping movements are abolished when these fish are additionally deprived of lateral line information.” [Bak-Coleman, J., Court, A., Paley, D. A., and Coombs, S. (2013). The spatiotemporal dynamics of rheotactic behavior depends on flow speed and available sensory information. Journal of Experimental Biology, 216(21), 4011-4024.]Furthermore, If you look at the V-/T- (bottom panel) of their Figure 10, Elder and Coombs show that cross-stream sweeping movements occur in LL+ fish in the centre of the test space – but with LL- fish the cross-stream movements occur at the inlet side of the space corresponding with the statement above and probably mediated by tactile sensing. “In particular, fish without visual cues exhibited cross-stream sweeping behaviors similar to those observed in visually-deprived giant danio (Devario aequipinnatus) (Bak-Coleman et al. 2013). This sweeping behavior could represent an active, compensatory strategy for, e.g., trying to make tactile contact with an upstream object to maintain position, or it could represent a passive loss in the ability to hold cross-stream position.” [Elder, J., and Coombs, S. (2015). The influence of turbulence on the sensory basis of rheotaxis. Journal of comparative physiology A, 201(7), 667-680.]So my view remains that this paper is an interesting modelling illustration of potential ‘invasive sensing’, but that it falls short of demonstrating that the biological evidence supports the case that fish actually do this. In a general sense, understanding animal orientation requires a physically plausible mechanism, but requires behavioural demonstration of the actual use of this in a biologically relevant way. The paper would be better framed to state the case for this possibility and layout the behavioural tests that would demonstrate (or not) its biological relevance.

Before explaining for the sake of an intellectual argument the reason why we believe the statements are flawed, let us acknowledge that in this revision we have further deemphasized statements on cross-sweeping movements, a phenomenon that is quite secondary to our main claims. Practically, the whole reference to this behavior is in a paragraph in the Discussion. Also, we have added text to inform new experimental research at the interface between biology and robotics.

Reviewer 1 is, in our view, cherry-picking statements without correctly connecting them to our model. The evidence they point to does not detract from the model, but rather it reinforces its predictive value. Let us start with the 2013 paper by Coombs and colleagues. The fact that abolishing the lateral line causes the experimental animals to lower cross-sweeping movements is in agreement with our model and not in contradiction. In fact, if you look at Figure 3a in our paper, the locations of the two equilibria with non-zero heading is varying with β/β^*^. The better the animal performs rheotaxis (β/β^*^ >>1), the closer the points are to the walls, thereby affording wider sweeps. For animals that have worse rheotaxis performance, β/β^*^ ~1 so that the two unstable equilibria will be closer to the origin and cross-sweeping movements will be reduced.

Moving to the 2015 paper by Elder and Coombs, the Reviewer is using the bottom panels of Figure 10 from this 2015 paper (V-/T-/LL+ with V-/T-/LL-) to argue that tactile sensing is the dominant mechanism underlying cross-sweeping movements for subjects with a compromised lateral line.

The circumstantial argument is based on the fact that the heat maps suggest that at high speed there is a tendency of fish with a compromised lateral line to swim more toward the inlet (right hand side). Such an argument has several flaws. First, the animals do not maintain continuous contact with the inlet and it is difficult from the heat maps to objectively argue about even a preference for the inlet. In fact, experiments in the 2013 paper above that detail the stream-wise position of animals for different flow conditions do not point at any difference in spatial preference along the flow direction. Figure 10 from the 2013 paper, shows that animals in the dark (black lines) do not change their streamwise position if their lateral line is compromised (right versus left) for any flow speed (zero, low, and high from top to bottom).

Second, the cross-stream position of these fish is closer the mid-section than those with intact lateral line, in agreement with our predictions (see the point above on the 2013 paper) that the range of cross-stream sweeps becomes wider for larger value of β/β*, that is, for as for increasing feedback from the lateral line. This evidence is even made clearer in Figure 8b from the same 2015 paper, which shows that V-/T-/LL+ has less crosstream positional variability than V-/T-/LL-. Finally, the companion figure, Figure 8a, shows other two key pieces of information that reinforce the idea that hydrodynamic coupling is a viable mechanism for cross-stream movements for animals deprived of vision: sweeps are periodic and their oscillation frequency increases with the flow speed, all in agreement with our model. The last point is explicitly reinforced by the claim made by the authors in the text: "there is a clear emergence of sweep behavior at flow speeds above 2 cm/s.… irrespective of the TGS condition or availability of lateral line information."

Hence, not only does this Reviewer fail to offer compelling evidence to dispute the model (all the presented evidence is in fact supporting our model), but also they focus all of their destructive efforts on a secondary element of the model, which we explicitly acknowledge as a topic that has yet to gather thorough and conclusive empirical observations (L312). We have constructed the entire validation effort on the rheotaxis threshold, for which we have inspected 300-400 papers and created a valuable dataset that can be used for validation. We do not fully understand the frustration of the Reviewer and their perceived hostility, maybe the root cause is our original reply to their comment #2 where we used an excessively forceful language, for which we apologize: "the claim of the Reviewer that the “rheotaxis thresholds in the Coombs etc. table most likely occur at a threshold for slip detection” is a conjecture. " Likely, this reply prompted such a passionate and rushed review of our work, which does not contain any of the manuscript beyond page 2.

Other comments:L 32 Do fish sense the flow and actively perform rheotaxis accordingly, or can rheotaxis be achieved passively? Biology often demonstrates redundancy so the answer to the above question is not an either/or.

We reworded the text accordingly.

L38 Several modern studies have unveiled the critical role of the lateral line system, an array of mechanosensory receptors located on the surface of fish body, in their ability to orient against a current hinting at a hydrodynamics-based rheotactic mechanism that has not been fully elucidated. I think that this is an interesting point – that adds significantly to the Oteiza et al.approach.

We agree that this is a key point of novelty and we have edited the text to make it clearer.

Reviewer #2 (Remarks to the Author):My concerns about this manuscript have been addressed by this revision by the authors.Reviewer #4 (Remarks to the Author):To investigate the fish orientation against a flow is a very interesting topic. We have noticed there are some recent studies combining hydrodynamics and biosensors. In a tube flow, it is shown that the rheotaxis behavior of fish is a significant phenomenon. In the present manuscript, the authors tried to illustrate that the rheotaxis exits even without a perceptual system, using a simple hydrodynamic model. This model introduced a bidirectional coupling between the motion of the fish (a dipole-like model) and the flow physics in its surroundings (walls and shearing flow). According to the stability analysis of this dynamic system, they found the critical speed and stability threshold reported in previous studies, and the cross-stream sweeping movements while swimming against the flow. This dynamic process can be independent of the feedback mechanism (based on the circulation measurement through the lateral line), which is a periodic solution. That means, the fish swimming against the flow in a tube is a hydrodynamic behavior, which will help the fish with weak or damaged biosensors swim in a specified tube.Some suggestions are listed as follows,Modify the title, including the keywords “tube” and “shear flow”.

We changed the title accordingly.

As a widely accepted conclusion, the rheotaxis behavior is the result of fish’s control by relying on the feedback of various sensors (vision, vestibular system, tactile sensors, lateral line system, etc.). The authors proposed a specific swimming scene that can reduce the dependence on sensors. It is necessary to rewrite the introduction, which will help the readers to understand the relationship between hydrodynamics and biology.

We edited the introduction to clarify the specific conditions we are focusing on.

About the invasive sensor, there is a lack of definition and deep discussion.

We added context throughout.